# Ability of prebiotic polysaccharides to activate a HIF1α-antimicrobial peptide axis determines liver injury risk in zebrafish

Zhen Zhang[1,3], Chao Ran[2,3], Qian-wen Ding[1], Hong-liang Liu[1], Ming-xu Xie[1], Ya-lin Yang[2], Ya-dong Xie[1], Chen-chen Gao[1], Hong-ling Zhang[1] & Zhi-gang Zhou[1]

Natural polysaccharides have received much attention for their ability to ameliorate hepatic steatosis induced by high-fat diet. However, the potential risks of their use have been less investigated. Here, we show that the exopolysaccharides (EPS) from *Lactobacillus rhamnosus* GG (LGG) and *L. casei* BL23 reduce hepatic steatosis in zebrafish fed a high-fat diet, while BL23 EPS, but not LGG EPS, induce liver inflammation and injury. This is due to the fact that BL23 EPS induces gut microbial dysbiosis, while LGG EPS promotes microbial homeostasis. We find that LGG EPS, but not BL23 EPS, can directly activate intestinal HIF1α, and increased HIF1α boosts local antimicrobial peptide expression to facilitate microbial homeostasis, explaining the distinct compositions of LGG EPS- and BL23 EPS-associated microbiota. Finally, we find that liver injury risk is not confined to *Lactobacillus*-derived EPS but extends to other types of commonly used natural polysaccharides, depending on their HIF1α activation efficiency.

[1] China-Norway Joint Lab on Fish Gut Microbiota, Feed Research Institute, Chinese Academy of Agricultural Sciences, 100081 Beijing, China. [2] Key Laboratory of Feed Biotechnology, Ministry of Agriculture and Rural Affairs, Feed Research Institute, Chinese Academy of Agricultural Sciences, 100081 Beijing, China. [3]These authors contributed equally: Zhen Zhang, Chao Ran. Correspondence and requests for materials should be addressed to Z.-g.Z. (email: zhouzhigang@caas.cn)

Non-alcoholic fatty liver disease (NAFLD), which is characterized by the accumulation of ectopic triacylglycerol in liver without excess alcohol consumption, has become the most common liver disorder worldwide, with a prevalence reaching 80–90% in obese adults in industrialized countries[1,2]. Obesity is a recognized risk factor for NAFLD[3]. Options for pharmacologic therapy that target NAFLD remain extremely limited[4,5]. In the search for potential pharmacological therapy, the beneficial effects of natural bioactive substances on hepatic steatosis are gaining increasing attention.

Recent work has indicated the importance of the gut-liver axis in the development of liver diseases, especially gut microbiota[6]. The gut microbiome contributes to NAFLD development[7,8]. Moreover, it determines the progression to non-alcoholic steatohepatitis (NASH)[9,10], which is characterized by hepatic inflammation and liver injury. Similarly, gut microbial dysbiosis is a key contributor to alcohol-induced steatohepatitis and liver injury[11,12]. Despite their importance, host factors responsible for the maintenance and regulation of intestinal homeostasis, particularly the microbiota, have been less investigated[13]. Hypoxia-inducible factor (HIF) has been recognized as an important regulator of intestinal homeostasis[14]. Previous studies mostly focused on the barrier-protective function of intestinal epithelial HIF. In this regard, intestinal HIF-1 has been proposed as a therapeutic target in colitis[15,16]. Recently, HIF-1α has been implicated to play a role in maintenance of intestinal homeostasis and the development of liver disease[17]. Deletion of intestinal HIF1α has been shown to worsen gut dysbiosis in response to alcohol exposure and exacerbate alcoholic liver disease[17].

Different natural compounds, such as berberine[18], silymarin[19,20], and eriocitrin[21], have been proven to attenuate hepatic steatosis. In particular, polysaccharides have received much attention recently as an anti-hepatic steatosis component[22,23]. Polysaccharides are polymeric carbohydrate macromolecules composed of long chains of monosaccharide units that are connected by various glycosidic linkages, and have a wide variety of biological activities[24]. Natural polysaccharides have shown appreciable effects on the amelioration of fatty liver disease, including polysaccharides from *Angelica sinensis*[25], Fuzhuan Brick Tea[26], and *Ginkgo biloba* leaf[27], *Ganoderma lucidum*[22], and *Pholiota nameko*[28]. In many cases, the polysaccharides exert functions as prebiotic agents, and conferr their anti-hepatosteatosis effects by microbiota modulation[22,26]. However, apart from the well-reported benefits, dysregulated fermentation of some prebiotic polysaccharides (fibers) was reported to induce microbial dysbiosis and hepatic inflammation and liver cancer in mice[29,30], implying risks associated with their application. Nevertheless, the underlying mechanisms determining the safety or risk of different polysaccharides when used as prebiotics are not clear.

Zebrafish (*Danio rerio*) is a useful vertebrate biomedical research model with favorable characteristics, i.e. high reproductive rate, transparent embryos and larvae, tractability in forward genetic screens, and genetic similarity to humans[31]. Moreover, there is high conservation at the anatomic and molecular levels of the hepatobiliary system in zebrafish compared with mammals, and several pharmacological, genetic and nutritional models of NAFLD have been established in zebrafish[32–34]. In this study, we established models of high-fat diet-induced hepatic steatosis in both adult and larval zebrafish. In light of the beneficial effects of *Lactobacillus* probiotics and the reported multi-functions associated with the *Lactobacillus* exopolysaccharides (EPS), the natural polysaccharides derived from bacteria, we tested the anti-hepatic steatosis capacity of the EPS from two well used probiotic *Lactobacillus* strains *Lactobacillus rhamnosus* GG (LGG) and *L. casei* BL23 in the zebrafish model.

The structure and monosaccharide composition of the two EPS's are different[35,36]. Our initial purpose was to test the anti-hepatosteatosis effect of the EPS. Intriguingly, although oral administration of both LGG EPS and BL23 EPS ameliorated hepatic steatosis in high-fat diet-fed zebrafish, BL23 EPS induced liver inflammation and injury. Moreover, the liver injury effect by BL23 EPS was attributable to dysbiosis of the intestinal microbiota, while LGG EPS improved the intestinal homeostasis. Further, we observed that the differentiation of the BL23 EPS and LGG EPS-associated microbiotas was mainly attributable to direct induction of HIF1a activation through TLR4ba by LGG EPS. In contrast, BL23 EPS was not able to induce HIF1a activation. Our results suggest there is potential risk in using natural polysaccharides to treat hepatic steatosis, and point to an important criterion in selecting safe polysaccharides for this purpose.

## Results

**High-fat diets induce hepatic steatosis in adult and larval zebrafish.** In mammals, high-fat diets have been used to induce hepatic steatosis[37–39]. Similarly, we formulated a high-fat diet for adult and larvae zebrafish (Supplementary Tables 1 and 2). We observed that high-fat diet feeding for 4 weeks led to a substantial increase in body weight and abdominal subcutaneous fat accumulation compared with a control group (Fig. 1a, b). Moreover, high-fat diet-induced higher triglyceride content in zebrafish liver versus control (Fig. 1c, d), indicating hepatic steatosis. Consistently, the expression of genes involved in lipogenesis in livers was upregulated by high-fat diet (Fig. 1e). Similarly, the livers of zebrafish larvae fed a high-fat diet for 7 exhibited more abundant lipid droplet accumulation versus control (Fig. 1f), with increased expression of lipogenesis genes (Fig. 1g). Taken together, these results demonstrate that high-fat diet leads to lipid metabolism disorders and hepatic steatosis in zebrafish.

**EPS from two *Lactobacillus* strains ameliorate hepatic steatosis.** EPS are ubiquitous components of the cell envelope of lactobacilli[40]. Our preliminary results indicated that LGG EPS and BL23 EPS can reduce triacylglyceride accumulation in zebrafish liver cells by inhibiting lipogenesis without inducing inflammation (Supplementary Fig. 1a–e). We further investigated whether oral administration of LGG EPS and BL23 EPS can reduce hepatic steatosis in high-fat diet-fed zebrafish. The results showed that LGG EPS and BL23 EPS decreased weight gain in high-fat diet-fed zebrafish at both 0.5 and 1.0% supplementation (Supplementary Fig. 2a). Moreover, lipid droplets accumulation and triacylglyceride levels were significantly reduced by LGG EPS and BL23 EPS at the two doses compared with the high-fat diet group (Fig. 2a, b). Feed intake did not vary significantly among control, high-fat diet, and the EPS supplemented high-fat diet groups, indicating that the effects of EPS on body weight and hepatosteatosis were not due to reduced feed consumption (Supplementary Fig. 2b). Similarly, LGG EPS and BL23 EPS decreased lipid droplet accumulation in liver of zebrafish larvae compared with the high-fat diet group (Fig. 2c, d). Consistent with the observed phenotypes, LGG EPS and BL23 EPS reverted the high-fat diet-associated expression of genes involved in lipogenesis and energy expenditure, to levels comparable to the control diet group (Fig. 2e–h).

**BL23 EPS induces liver inflammation and injury.** Compared to the high-fat diet group, LGG EPS-fed zebrafish were characterized by significantly lower serum alanine aminotransferase (ALT) and aspartate aminotransferase (AST) activity (Fig. 3a, b). Inflammation was also decreased in the liver of LGG EPS-fed zebrafish (Fig. 3c). Intriguingly, although BL23 EPS also reduced

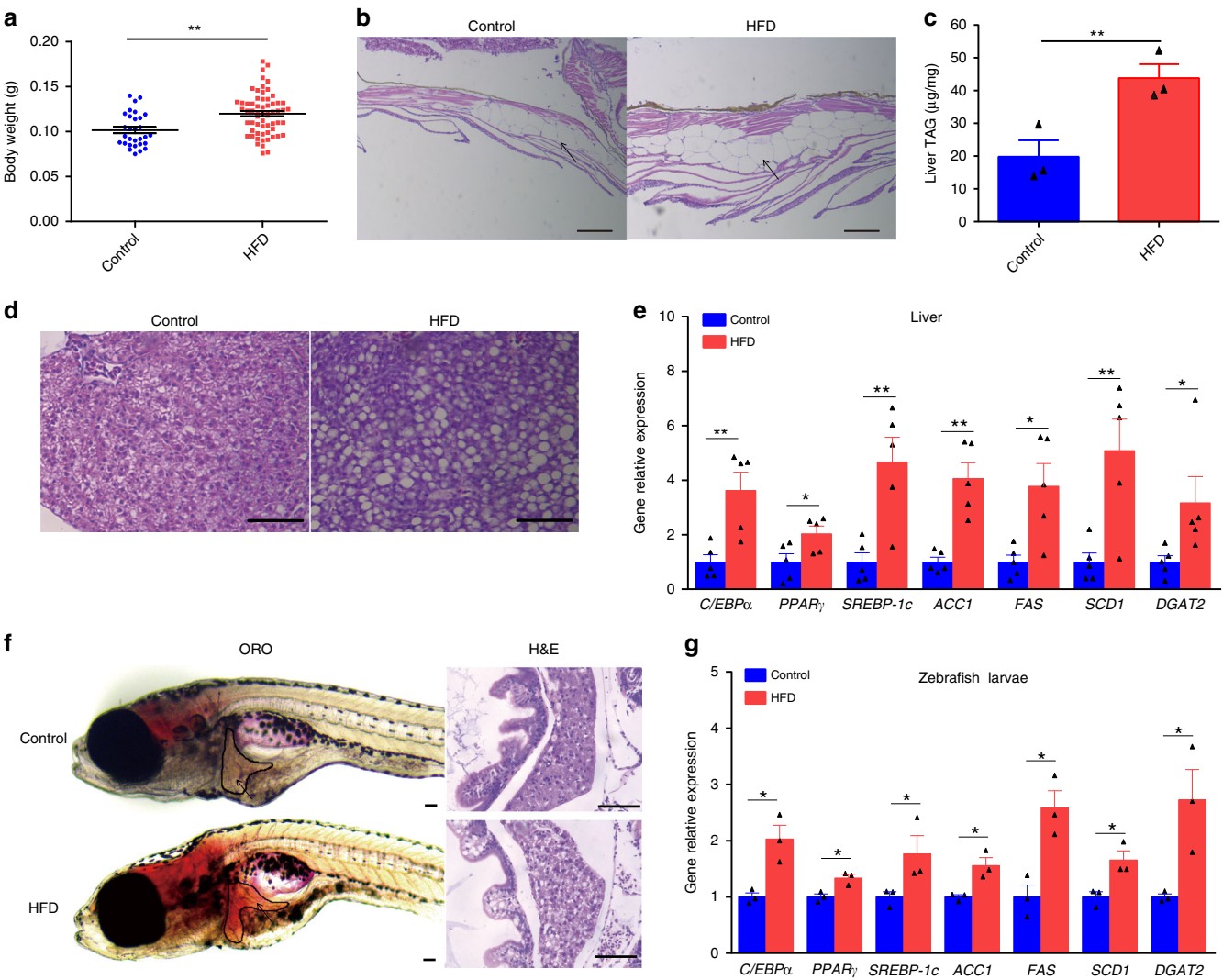

**Fig. 1** High-fat diets induced hepatic steatosis in adult and larval zebrafish. **a** Body weight of adult zebrafish fed with a control diet or high-fat diet (HFD) for 4 weeks. **b** Representative image of the abdominal subcutaneous fatty tissue histology with H&E staining. The scale bar is 100 μm. **c** Triacylglyceride (TAG) content in liver (n = 3, pool of three zebrafish per sample). **d** Representative liver histology image by H&E staining. The scale bar is 50 μm. **e** The expression of lipid metabolism-related genes in liver (n = 5, pool of three zebrafish per sample). **f** Representative image of whole-mount oil red O staining and H&E staining of liver sections in larvae fed control diet and high-fat diet. The scale bar is 100 μm. **g** The expression of lipid metabolism-related genes in zebrafish larvae. Gene expression was analyzed using cDNA prepared from pools of larvae in each group after seven days of feeding (n = 3, pool of 20 larvae per sample). Significance was established using a two-tailed student's t-test. Differences are considered significant at P < 0.05 (*) and P < 0.01 (**)

hepatic steatosis as did LGG EPS, the fish fed a BL23 EPS sup-plemented diet had significantly higher serum ALT and AST relative to the high-fat diet group (Fig. 3a, b), and the hepatic inflammation was worsened (Fig. 3c). Moreover, BL23 EPS led to hepatocyte apoptosis, as shown by TUNEL staining and the expression of pro-apoptotic and anti-apoptotic factors (Fig. 3d, e). Together, these results indicate that although both EPS's ame-liorated hepatic steatosis in zebrafish fed a high-fat diet, BL23 EPS led to liver inflammation and injury, while LGG EPS showed liver protective effect.

**The liver injury effect of BL23 EPS is mediated by the intestinal microbiome**. To investigate the underlying mechanism for liver injury by BL23 EPS, we treated germ-free zebrafish with BL23 EPS and LGG EPS. Neither EPS induced an inflammatory response (Fig. 4a), and no apoptosis was observed by TUNEL staining in BL23 EPS-treated germ-free fish (Fig. 4b), indicating that the liver injury by BL23 EPS was not induced by the EPS per se.

Previous studies have reported a link between gut microbial dysbiosis and liver injury[11,12,41]. We therefore tested whether the BL23 EPS-induced liver injury was mediated by the microbiome. We observed that BL23 EPS, but not LGG EPS, led to intestinal bacterial outgrowth (Fig. 4c). The microbiome composition was assessed by Illumina sequencing across the V3–V4 regions of the 16S rRNA gene. Most of the identified reads from the digesta samples of all the four treatments belonged to two phyla (Fusobacteria and Proteobacteria) (Fig. 4d). Both LGG EPS and BL23 EPS altered the microbiota compared with the high-fat diet group (Supplementary Fig. 3b, c; Supplementary Table 3). However, the BL23 EPS microbiome featured increased abundance of Proteobacteria and reduced abundance of Fusobacteria compared with the high-fat diet group, while the LGG EPS microbiome revealed an opposite differentiation relative to BL23 EPS (Supplementary Table 4). At the genus level, *Cetobacterium* and *Plesiomonas* are the most discriminatory phylotypes differentiating the BL23 EPS- and LGG EPS-associated microbiotas, which belong to Fusobacteria

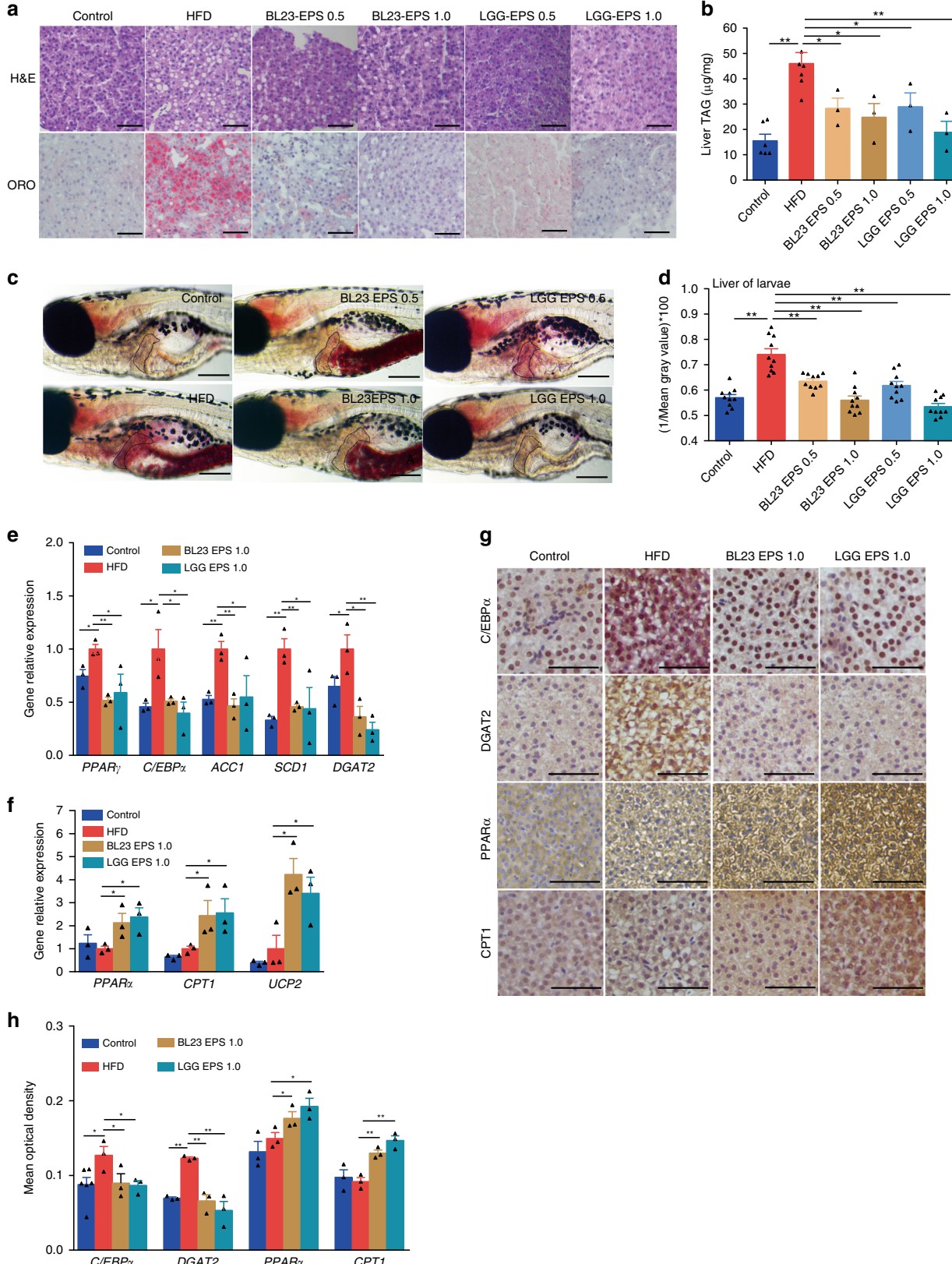

and Proteobacteria, respectively (Supplementary Table 5, Supplementary Fig. 3a). We next sought to investigate whether the liver injury in BL23 EPS-treated zebrafish was driven by the altered microbiota. We treated BL23 EPS-fed zebrafish with the nonabsorbable antibiotics Polymyxin B and Neomycin to deplete the commensal microbes. We observed that antibiotic

treatment significantly reduced the serum ALT and AST levels in BL23 EPS-fed zebrafish (Fig. 4e, f), indicating that BL23 EPS-altered microbiota was a major contributor to the liver injury. To confirm these results, we transferred intestinal microbiota of adult zebrafish fed control, high-fat diet, or high-fat diet containing LGG EPS or BL23 EPS for 4 weeks to germ-free

**Fig. 2** LGG EPS and BL23 EPS ameliorated the hepatic steatosis in high-fat diet-fed zebrafish. Adult zebrafish (1-month-old) were fed with the control diet, HF diet, or HF diet supplemented with 0.5% or 1% EPS for 4 weeks. **a** Representative liver histology images with oil red O staining and H&E staining. The scale bar is 50 μm. **b** Triacylglyceride contents in the liver ($n = 3$ or 6, pool of three zebrafish per sample). **c** Representative images of whole-mount oil red O staining of larvae fed control diet, high-fat diet, or high-fat diet supplemented with 0.5 or 1% EPS for one week. The scale bar is 200 μm. **d** Quantitative evaluation of hepatic steatosis in zebrafish larvae fed control diet, high-fat diet, or high-fat diet supplemented with 0.5 or 1% EPS for one week ($n = 10$). The ORO images in panel **c** were converted to 8-bit gray scale, and mean gray value was measured using ImageJ software to quantitatively evaluate hepatic steatosis. The expression of genes related to lipogenesis (**e**), energy expenditure (**f**), in livers as measured by $q$-PCR ($n = 3$, pool of three zebrafish per sample). **g**, **h** Immunohistochemical analysis and quantification of hepatic C/EBPα, DGAT2, PPARα, and CPT1 levels, the scale bar is 50 μm. Data were expressed as the mean ± SEM. Differences are considered significant at $P < 0.05$ (*) and $P < 0.01$ (**)

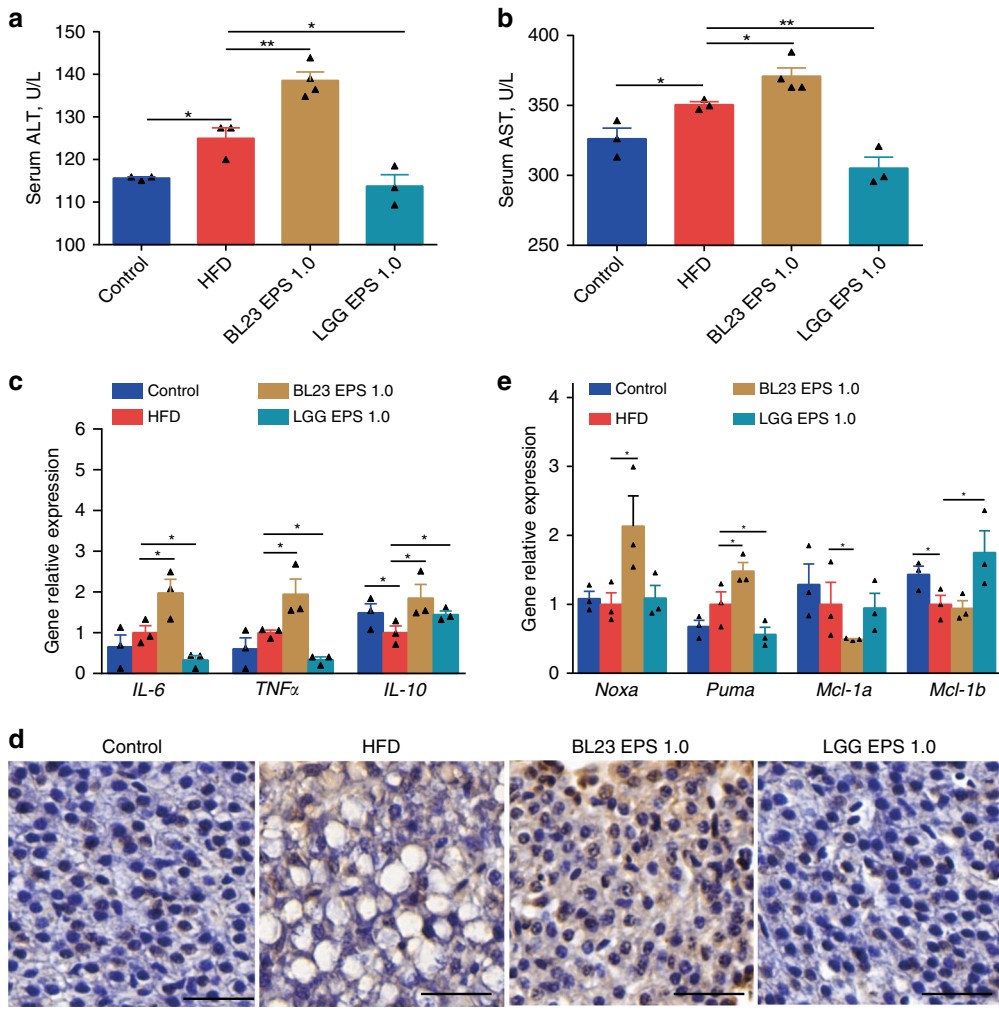

**Fig. 3** BL23 EPS induced liver inflammation and injury in high-fat diet-fed zebrafish. Adult zebrafish (1-month-old) were fed with the control diet, high-fat diet (HFD), or HFD supplemented with 1% EPS for 4 weeks. Serum ALT (**a**) and serum AST (**b**) in zebrafish. The expression of genes related to inflammation (**c**), pro-apoptotic factors (*Noxa* and *Puma*) and anti-apoptotic factors (*Mcl-1a* and *Mcl-1b*) (**e**) in livers as measured by $q$-PCR ($n = 3$, pool of three zebrafish per sample). **d** Representative liver histology images with TUNEL staining. The scale bar is 30 μm. Data were expressed as the mean ± SEM. Differences are considered significant at $P < 0.05$ (*) and $P < 0.01$ (**)

zebrafish, and tested the ALT and AST levels of the zebrafish larvae after eliminating the viscera. The results showed that the ALT and AST phenotypes in germ-free recipients colonized with microbiota from different groups recapitulated those observed in the donor zebrafish. LGG EPS microbiota was associated with lower ALT and AST levels compared with high-fat diet microbiota, while BL23 EPS microbiota induced higher ALT and AST levels (Fig. 4g, h). Together, these results supported the hypothesis that the liver injury effect of BL23 EPS was driven by the microbiome alteration. Consistent with the liver injury phenotype, the microbiota transfer results showed that LGG EPS microbiota can reduce inflammation, while BL23 EPS microbiota enhanced inflammation in the colonized zebrafish (Fig. 4i). Further, we colonized germ-free zebrafish with representative *Cetobacterium* and *Plesiomonas* strains isolated from the intestinal content of adult zebrafish. We observed that the *Plesiomonas* strain led to significantly enhanced inflammation compared with the *Cetobacterium* strain (Fig. 4j), suggesting that their differential abundance in BL23 EPS and LGG EPS microbiomes may contribute to the overall inflammation phenotype associated with the microbiota.

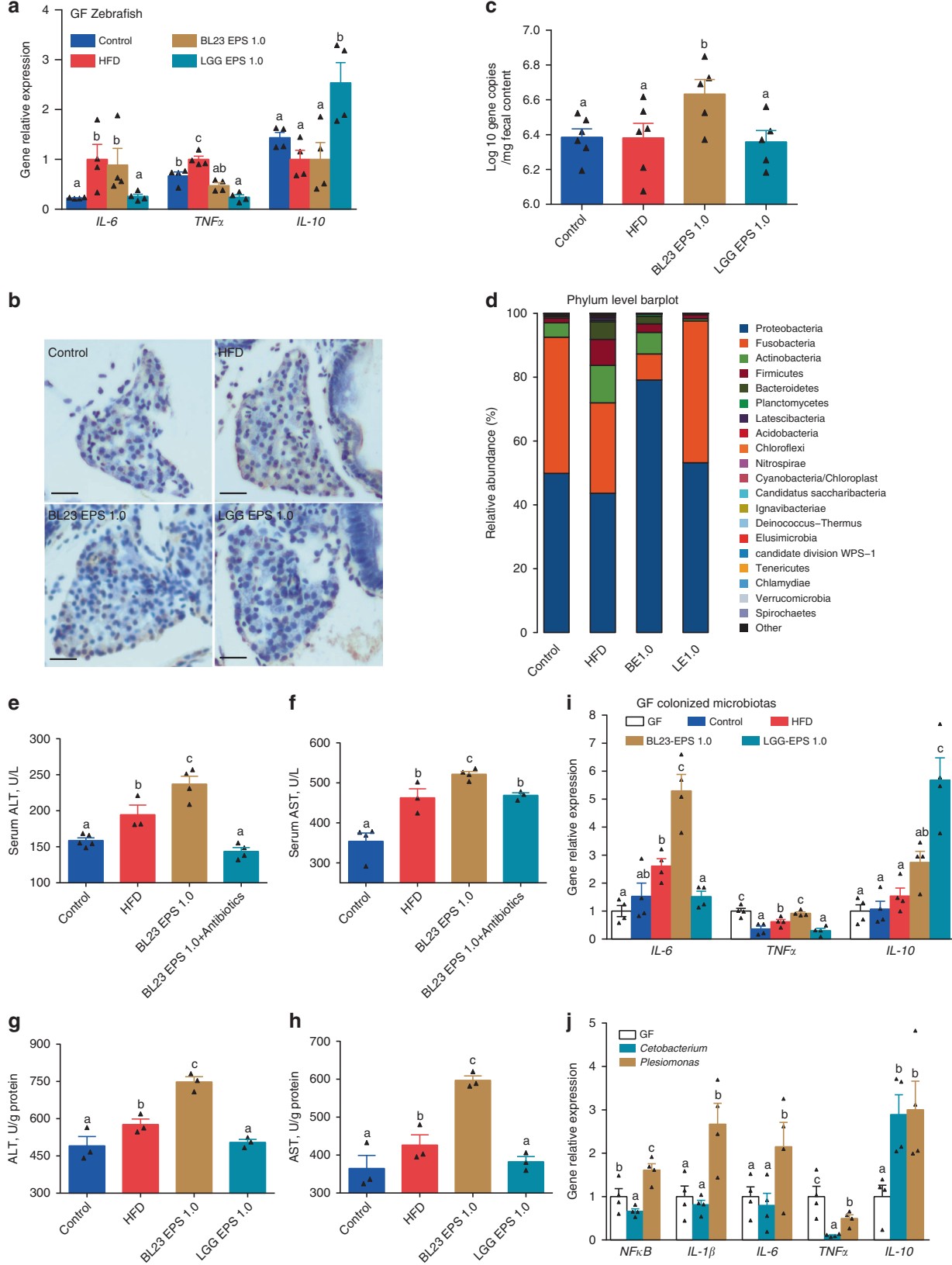

These observations suggest that bacterial products translocated form BL23 EPS-altered microbiota promote a signaling cascade in the liver and induce inflammation and injury. We therefore tested the expression of important pattern recognition receptor (PRR) genes in the liver of BL23 EPS-treated zebrafish, and observed enhanced expression of *TLR1* and *LBP* (Fig. 5a–c). BL23 EPS can directly induce expression of *TLR1* in germ-free zebrafish, with no induction of inflammation and injury (Supplementary Fig. 3d–f), suggesting that the TLR1 signaling in the liver of BL23 EPS-treated fish was not

**Fig. 4** The liver injury effect of BL23 EPS is mediated by the gut microbial dysbiosis. **a** The expression of genes related to inflammation as measured by q-PCR in germ-free zebrafish larvae fed with sterile control diet, high-fat diet (HFD), or HFD supplemented with 1.0% BL23 EPS or 1.0% LGG EPS for one week ($n = 4$, pool of 20 larvae per sample). **b** Liver TUNEL staining in germ-free (GF) larvae fed with sterile diets for one week. The scale bar is 50 μm. **c** Total number of bacteria (log 16S rRNA gene copies/mg intestinal contents) in adult zebrafish fed with the control diet, HF diet, or HF diet supplemented with 1.0% BL23 EPS or 1.0% BL23 EPS for 4 weeks. **d** The relative phyla abundance of the microbiota of adult zebrafish fed with different diets for 4 weeks. Serum ALT (**e**) and serum AST (**f**) in adult zebrafsh fed with the control diet, HF diet, 1.0% BL23 EPS diet or 1.0% BL23 EPS supplemented with nonabsorbable antibiotics mixture (0.25% Polymyxin B and 0.33% Neomycin) for two weeks. ALT levels (**g**) and AST levels (**h**) in GF larvae colonized with four different gut microbiotas, and fed sterile HFD diets for seven days ($n = 4$, pool of 20 larvae per sample). ALT and AST levels were assayed in zebrafish after eliminating the viscera. **i** The expression of genes related to inflammation as measured by q-PCR in GF larvae colonized with four different gut microbiotas, and fed sterile HFD diets for seven days ($n = 4$, pool of 20 larvae per sample). **j** The expression of genes related to inflammation as measured by q-PCR in GF larvae fed sterile HFD diets for seven days, and colonized with *Cetobacterium* YZ1 or *Plesiomonas* YZ2 for another 24 h ($n = 4$, pool of 20 larvae per sample). Data are expressed as the mean ± SEM. Graph bars labeled with different letters represent statistically significant results ($P < 0.05$), whereas bars with the same letter indicates non-significant differences

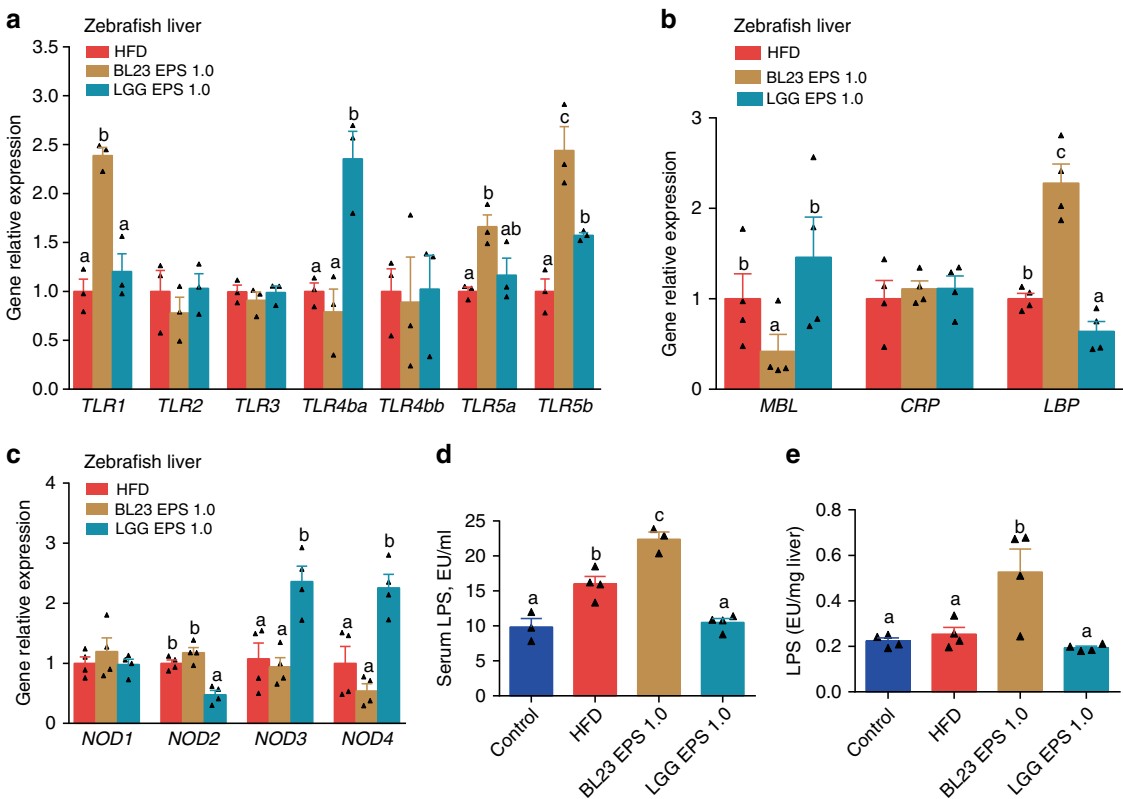

**Fig. 5** The liver injury effect of BL23 EPS involved the translocation of intestinal LPS. Adult zebrafish (1-month-old) were fed with control diet, high-fat diet (HFD), or HFD supplemented with 1.0%BL23 EPS or 1.0%LGG EPS for 4 weeks. The hepatic expression of TLRs genes (**a**), secreted PRRs (MBL, CRP, LBP) genes (**b**), and cytosolic PRRs (NODs) genes (**c**) in zebrafish livers as measured by q-PCR ($n = 3$ or 4, pool of three zebrafish per sample). Serum LPS (**d**) and hepatic LPS (**e**) in zebrafsh ($n = 3$ or 4, pool of three zebrafish per sample). Data are expressed as the mean ± SEM. Graph bars labelled with different letters on top represent statistically significant results ($P < 0.05$), whereas bars with the same letter correspond to results that show no statistically significant differences

triggered by transplanted bacterial products. Notably, *LBP* expression was not induced by BL23 EPS treatment in either zebrafish liver cells or germ-free zebrafish (Supplementary Fig. 3d–f), suggesting that LBP was induced by transplanted microbial products from the BL23 EPS-altered microbiota. LBP is a soluble receptor that binds the endotoxin lipopolysaccharide (LPS). This suggests that the LPS translocated from intestinal lumen contributed to the liver injury effect of BL23 EPS. Consistently, we observed that the LPS level in the serum and liver of BL23 EPS-fed zebrafish was significantly higher than in high-fat diet fish (Fig. 5d, e). This is consistent with the expansion of Proteobacteria in the BL23 EPS-altered micro-biota, as the phylotypes in this phylum are major LPS producers.

**Differentiation of LGG and BL23 EPS-associated microbiomes involves a HIF1α-AMP axis.** The results above indicated that the BL23 EPS-altered microbiota, which featured enrichment of Proteobacteria and reduction of Fusobacteria, contributed to liver inflammation and injury. We next investigated the mechanism underlying the opposite differentiation of microbiota associated with BL23 EPS versus LGG EPS. We firstly supplemented high-fat diet with the combination of monosaccharides composed of each EPS, with the amount and molar ratio the same as the corresponding polysaccharides[35,36]. We observed that after feeding for 3 weeks, the monosaccharide combination of either EPS was not able to reduce liver triacylglyceride (Fig. 6a), and led to increased serum ALT/AST and LPS, to levels comparable to that of the BL23 EPS group (Fig. 6b–d). Moreover, the monosaccharide

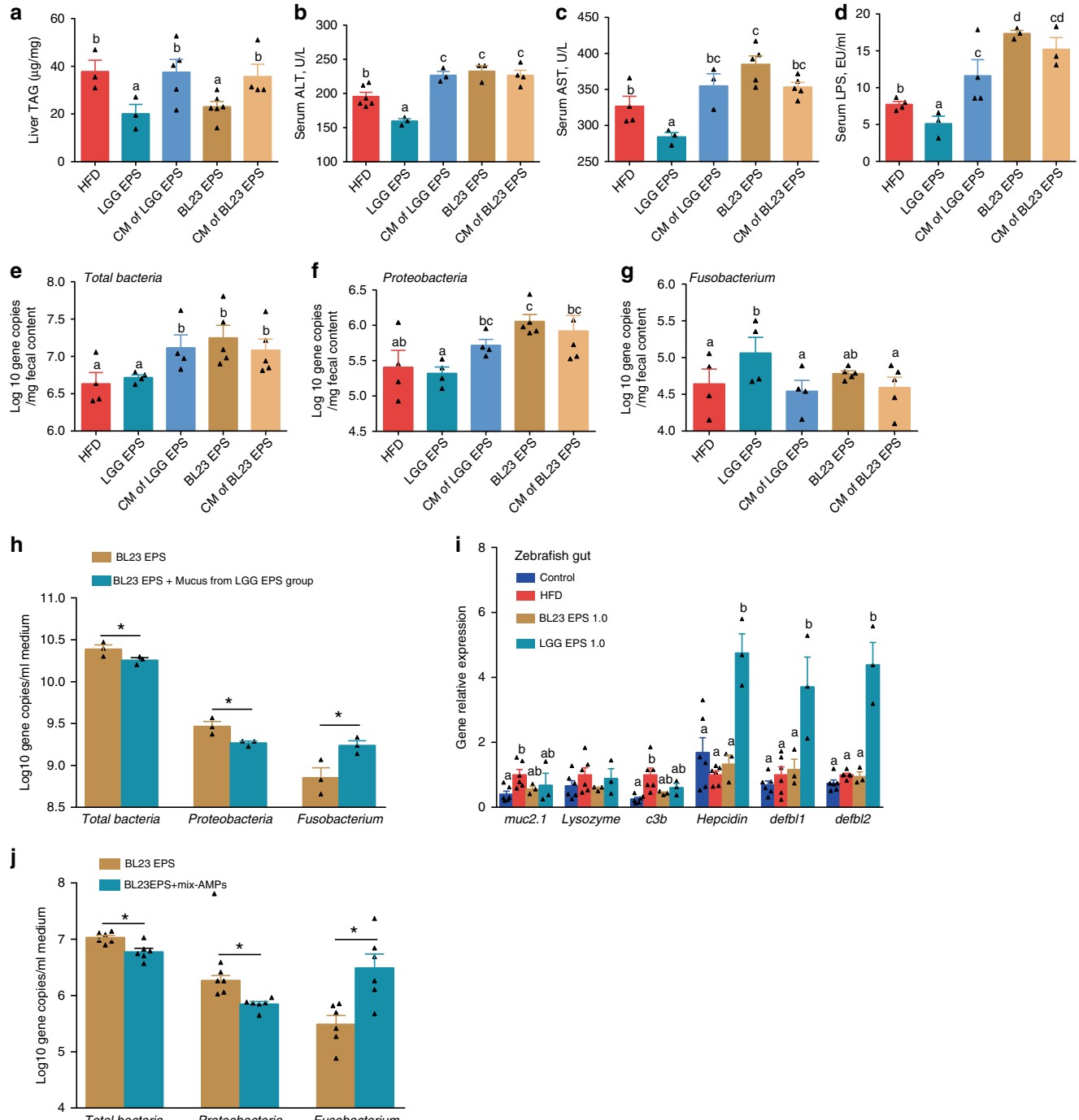

**Fig. 6** The microbial homeostasis associated with LGG EPS is mediated by the induction of AMPs. Adult zebrafish were fed with the HF diet, 1.0 %LGG EPS diet, 1.0 %BL23 EPS or HF diet supplemented with 1.0% combination of monosaccharides comprised of LGG EPS or BL23 EPS at 1.0% for 4weeks. **a** Triacylglyceride (TAG) contents in the liver. Serum ALT (**b**), serum AST (**c**), and serum LPS (**d**) in zebrafsh. Total number of bacteria (log 16S rRNA gene copies/mg intestinal contents) (**e**), the number of Proteobacteria (**f**), and Fusobacteria (**g**) in the intestinal microbiota. The BL23 EPS- microbiota from adult zebrafish were cultured for 12 h in M9 (minimal medium; Fluka) supplemented with 0.2% BL23 EPS or 0.2% BL23 EPS+ mucus from LGG EPS treated adult zebrafish, respectively. The number of total bacteria (log 16S rRNA gene copies/ml medium), Proteobacteria, and Fusobacteria after in vitro culture (**h**). **i** The expression of genes related to intestinal secretions in the intestines as measured by *q*-PCR ($n = 6$, pool of three zebrafish per sample). Synthetic mix-AMPs (100 ng Hepcidin, 100 ng β-defensin1, and 100 ng β-defensin2) (Top-peptide, Shanghai) was administered by single intraperitoneal injection to adult zebrafish fed 1.0% BL23 EPS diet, and zebrafish were fed another 3 days, and the bacterial numbers were evaluated (**j**). Data are expressed as the mean ± SEM. Graph bars labeled with different letters on top represent statistically significant results ($P < 0.05$), whereas bars with the same letter indicates non-significant differences. HFD, high-fat diet

combination of both EPS's altered the microbiota in a way that resembled that of BL23 EPS, which featured Proteobacteria enrichment and Fusobacteria reduction (Fig. 6e–g). Additionally, the monosaccharide combination of both EPS's led to bacterial overgrowth, similar to what was observed with BL23 EPS. These results indicate that the differentiation of LGG EPS and BL23 EPS-associated microbiomes was not due to differences in the fermentation substrates for the gut bacteria. We therefore hypothesized that the LGG EPS can directly interact with the host, which depends on its polysaccharide structure integrity, and the interaction induces some host signaling that maintains microbial homeostasis.

We next tested whether the mucus from LGG EPS-treated zebrafish contributed to the microbial homeostasis associated with the EPS. We observed that administration of LGG EPS-associated mucus to in vitro culture of BL23 EPS-associated microbiome led to a shift in the microbial composition toward the configuration of LGG EPS microbiota (Fig. 6h), suggesting that intestinal secretions of the LGG EPS group altered the gut microbiota. We further found LGG EPS significantly induced the expression of genes related to antimicrobial peptides (AMPs) compared with either the high-fat diet or BL23 EPS groups. AMPs were not induced in response to mucin, lysozyme, or complement component (Fig. 6i). AMPs have been reported to regulate microbial composition[42–44]. To investigate whether the increase of AMP levels was responsible for the microbiota-reverting effect of LGG EPS-associated mucus, we injected a mixture of synthetic AMPs, including defb1, defb2, and hepcidin[45], to BL23 EPS-fed zebrafish. Intriguingly, injection of AMPs changed the BL23 EPS-microbiota toward features associated with LGG EPS (Fig. 6j), supporting our hypothesis that the AMPs are the microbiome-modulating effectors responsible for the microbial homeostasis associated with LGG EPS.

**HIF-1α mediates the effect of LGG EPS on AMP production.** HIF is a master transcription factor regulating a variety of genes in the intestine, and antimicrobial peptides have been reported to be positively regulated by HIF-1α[46,47]. We therefore assessed the HIF1α expression in the intestine of zebrafish fed different diets. We observed significantly higher expression of HIF1α in zebrafish fed LGG EPS, compared with the BL23 EPS and high-fat diet groups (Fig. 7a–c). To investigate the potential association between HIF1α signaling and the differentiation of LGG EPS- and BL23 EPS-microbiota, we took advantage of the foxo3b[−/−] zebrafish, which has enhanced HIF activity[48]. We fed wild-type and foxo3b[−/−] zebrafish larvae BL23 EPS-supplemented diet for 1 week. Compared with wild-type, the intestinal microbiota of foxo3b[−/−] zebrafish featured marked compositional difference which resembled that associated with LGG EPS, as manifested by the reduction in bacterial overgrowth (Fig. 7d), the reduction of Proteobacteria/Plesiomonas, and enrichment of Fusobacteria/Cetobacterium (Fig. 7e–h). Accordingly, the AMP levels were significantly increased in the foxo3b[−/−] zebrafish versus wild-type (Fig. 7i).

To further confirm the involvement of HIF1α, we injected YC-1, an inhibitor of HIF, to LGG EPS-fed zebrafish. The enhancement of HIF expression by LGG EPS was reverted by YC-1 treatment (Fig. 7j), and the AMP levels were significantly decreased by the compound (Fig. 7k). Interestingly, the LGG EPS microbiota shifted toward that associated with BL23 EPS after YC-1 treatment, with bacterial overgrowth and an expansion of Proteobacteria (Fig. 7l). Moreover, YC-1 treatment elevated the level of ALT (Fig. 7m) and AST (Fig. 7n) and increased the expression of pro-apoptotic factors (Noxa and Puma) (Fig. 7o) in

livers of the LGG EPS supplemented zebrafish. Together, these results indicated that the differentiation of LGG EPS and BL23 EPS microbiotas involved the HIF1α-AMP axis. While both EPS's have the potential to lead to microbial dysbiosis via fermentation substrates by the gut bacteria in a high-fat diet-fed fish, LGG EPS can directly interact with the host and stimulate the expression of HIF1α (Fig. 7p), which induces AMPs and maintains microbial homeostasis. The BL23 EPS does not have the HIF1α-activation effect due to structural differences in the polysaccharide, and leads to microbial dysbiosis characterized by an expansion of Proteobacteria and liver injury.

**Structural basis for the interaction between EPS and the receptors.** We next sought to investigate how LGG EPS activated HIF1α. We tested the expression of PRRs in the intestine of zebrafish fed an LGG EPS diet. The expression levels of TLR4ba, NOD3, and NOD4 were increased (Fig. 8a). Previous studies indicated that TLR4 signaling can regulate the expression of HIF1[49,50]. Therefore, we investigated the role of TLR4ba in LGG EPS induced HIF1α stimulation.

The expression of TLR4ba was decreased by treating germ-free zebrafish with an in vivo antisense TLR4ba morpholino or with a control morpholino. Figure 8b shows a substantial reduction in TLR4 expression with vivo antisense morpholino oligomers. Knocking down TLR4ba inhibited the induction of Hif1αa and Hif1αb by LGG EPS (Fig. 8c). These results indicated that the activation of HIF1α by LGG EPS was mediated by TLR4.

To study the structural determinants in LGG EPS responsible for TLR4 recognition, LGG EPS was processed with β-(1 → 3,4,6)-galactosidase (Sigma, G1288) to remove the branches (Fig. 8d). We treated germ-free zebrafish with enzyme-treated EPS, and observed that the treatment reduced LGG EPS-induced expression of NOD3 and NOD4, with no effect on the expression of TLR4ba (Fig. 8e). Accordingly, enzyme-treated LGG EPS maintained its ability to activate HIF1αa and HIF1αb (Fig. 8f). These results indicate that the main strand of LGG EPS was responsible for TLR4ba interaction and HIF1 activation.

**Butyrate contributes to differentiation of the EPS-associated microbiota.** Short chain fatty acids (SCFA) are important metabolites of the microbiota. We tested the SCFA levels produced by BL23 EPS and LGG EPS-associated microbiota. Both BL23 EPS and LGG EPS increased the acetate and propionate in the intestinal content compared with the high-fat diet group (Fig. 9a, b). However, the butyrate and isobutyrate levels were only increased by LGG EPS, with a reduction seen in the BL23 EPS group compared with the high-fat diet group (Fig. 9c, d). Consistently, in vitro tests indicated that the representative Cetobacterium produces significantly more butyrate and isobutyrate that the representative Plesiomonas strain (Supplementary Fig. 4a–d). Butyrate has been reported to stimulate epithelial metabolism and deplete $O_2$, resulting in lower $O_2$ in the intestinal microenvironment[51]. Consistent with this correlation, we observed reduced partial oxygen pressure in the gut microenvironment of LGG EPS-treated zebrafish, while the intestinal mucosa of BL23 EPS was associated with $O_2$ levels higher than the high-fat diet group (Fig. 9e, f, Supplementary Fig. 4e–h). As hypoxia may stabilize HIF1α, the higher butyrate production by the LGG EPS-associated microbiota may further contribute to microbial homeostasis, while the low butyrate associated with BL23 EPS supports the dysbiotic state.

**The liver injury risk of commonly used polysaccharides.** To assess the liver injury risk of other commonly used prebiotic polysaccharides, we collected nine natural polysaccharides with

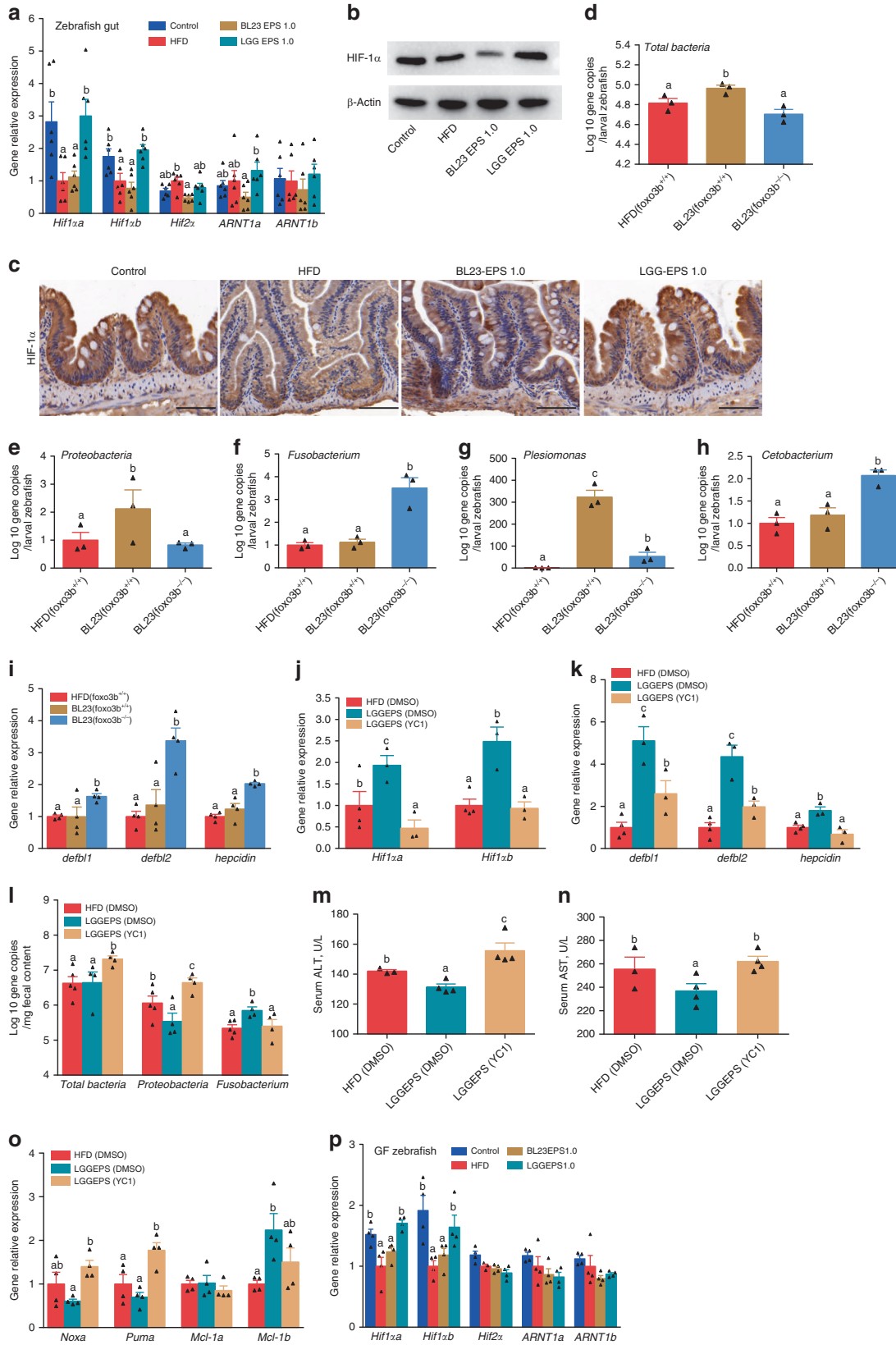

reported anti-steatosis effects[22,25,26,29,52–56]. High-fat diet-fed germ-free zebrafish were treated with nine other natural polysaccharides besides LGG EPS and BL23 EPS, and the expression of HIF1α in each group was tested. Notably, the polysaccharides can be categorized into two groups based on their HIF1α

activation efficiency. Four polysaccharides (from *Ganoderma lucidum*, *Angelica sinensis*, *Hirsutella sinensis*, and *Laminaria digitata*) enhanced the expression of HIF1α similarly to LGG EPS, while the other five polysaccharides (from *Camellia sinensis*, Guar gum, *Lycium chinense*, *Panax ginseng*, and *Astragalus*

**Fig. 7** HIF1α-AMP axis contributed to the microbial homeostasis associated with LGG EPS in high-fat diet-fed zebrafish. **a** The expression of genes related to hypoxia-inducible factors in the intestines of zebrafish fed with different diets for 4 weeks as measured by q-PCR ($n = 6$, pool of three zebrafish per sample). **b** A representative western blotting showing the expression pattern of Hif1α in the intestines. **c** Intestinal Hif1α levels evaluated by immunohistochemical analysis, the scale bar is 50 μm. Foxo3b-null zebrafish larvae (foxo3b$^{-/-}$) and their wild-type siblings (foxo3b$^{+/+}$) were fed 1.0% BL23 EPS diet for 7 days. The total bacteria (**d**) and the relative bacterial abundance of Proteobacteria (**e**), Fusobacteria (**f**), Plesiomonas (**g**), Cetobacterium (**h**) of the microbiota of the foxo3b-null larvae and their wild-type siblings ($n = 3$, pool of 20 larvae per sample). The expression pattern of genes related to AMPs **i** in foxo3b-null larvae and their wild-type siblings as measured by q-PCR ($n = 4$, pool of 20 larvae per sample). The adult zebrafish fed 1.0% LGG EPS for 2 weeks received intraperitoneal injection of YC-1 (an HIF-1α inhibitor; APExBIO, B7641) in dimethyl sulfoxide (DMSO) (2 mg/kg) once every 2 days for another 6 days. The expression of HIF-1α (**j**), and AMPs (**k**), in intestines as measured by q-PCR ($n = 3$ or 4, pool of three zebrafish per sample). **l** The number of total bacteria, Proteobacteria, and Fusobacteria in intestinal contents of zebrafish ($n = 4$ or 5). Serum ALT (**m**) and serum AST (**n**) in zebrafish. **o** The expression of apoptotic factors in intestines as measured by q-PCR ($n = 4$–6, pool of three zebrafish per sample). **p** The expression of genes related to hypoxia-inducible factors as measured by q-PCR in germ-free larvae fed with sterile control diet, high-fat diet (HFD), or HFD supplemented with 1.0% BL23 EPS or LGG EPS for 1 week ($n = 4$, pool of 20 larvae per sample). Data are expressed as the mean ± SEM. Graph bars labeled with different letters on top represent statistically significant results ($P < 0.05$), whereas bars with the same letter indicates non-significant differences

membranaceus) showed no HIF1α activation effect (Fig. 10a). We then tested the liver injury risk of the polysaccharides in adult zebrafish. Intriguingly, the five polysaccharides with no HIF1α activation activity all led to increased serum AST levels after 4 weeks of feeding (significant difference for CS-, GG-, and LC-polysaccharides; numerical difference for PG- and AM-polysaccharides), indicating liver injury risk (Fig. 10b). In contrast, the other four polysaccharides that caused HIF1α activation led to unchanged (for GL-polysaccharides) or significantly decreased levels of serum AST (for the other three polysaccharides) compared with the high-fat diet group (Fig. 10b). These results suggested that the liver injury risk observed in our study may be extrapolated to other commonly used prebiotic polysaccharides, and it is correlated with the HIF1α activation efficiency of the polysaccharides.

## Discussion

In this study, we observed unexpected liver injury by BL23 EPS, despite its ability to ameliorate hepatic steatosis in zebrafish fed a high-fat diet. In contrast, the LGG EPS improved liver health as indicated by lowered inflammation and serum ALT/AST levels. We further uncovered the mechanisms underlying the differential phenotypes associated with the two polysaccharides, which involved opposite differentiation of microbiota and the HIF1α-AMP axis.

The liver inflammation and injury in the BL23-EPS group resembles that of other liver diseases, such as alcoholic liver disease and NASH, except that steatosis was reversed by the EPS treatment. For both alcoholic liver disease and NASH, the liver pathology has been associated with gut dysbiosis, and intestinal decontamination with antibiotics has been reported to reduce alcoholic liver disease in mice, indicating an causal relationship between microbial dysbiosis and liver pathology[57]. Gut microbial products translocated from the intestinal lumen play important roles in the liver pathology. The development of alcoholic liver disease involves gut-derived LPS and TLR4-mediated signaling in the liver[58]. Similarly, hepatic TLR2, TLR4, and TLR9 signaling has been reported to contribute to the pathogenesis of NASH, with the gut bacterial peptidoglycan, LPS and CpG-containing DNA as the corresponding ligand, respectively[59–61]. In our study, we also observed that liver inflammation and injury was attributable to the dysbiosis in the intestinal microbiota, as supported by both antibiotic treatment and microbiota-transfer experiments. BL23-EPS-treated zebrafish featured enhanced serum LPS, which was consistent with induced hepatic expression of lipopolysaccharide-binding protein. This indicates that the gut bacterial LPS translocated from the intestinal lumen contributed to the liver injury. Moreover, we also observed elevated TLR5

expression in the liver of BL23-EPS-treated fish, implying that the TLR5 agonists from the gut microbiota might also be a contributor to the liver pathology induced by BL23-EPS. Notably, hepatic inflammation and injury was accompanied by steatosis in both alcoholic liver disease and NASH. BL23-EPS improved hepatic steatosis but induced inflammation, which are seemingly contrasting effects. Similarly with BL23-EPS, guar gum also induced liver inflammation and injury while improving hepatic steatosis[29]. These results indicate that hepatic inflammation and injury may occur independently of steatosis in some conditions, suggesting a multifactorial etiology of this symptom.

Hypoxia is an important factor for HIF1α protein stabilization[14]. On top of this, studies have shown that a number of nonhypoxic stimuli increase HIF-1 in a cell-specific manner, and some of these increases are equal or greater to that of hypoxic induction[62]. However, nonhypoxic induction of HIF-1 in intestinal epithelial cells has been rarely reported. The key finding of this study is that the LGG-EPS can directly induce the expression of intestinal HIF1α, which maintains microbial homeostasis. To our knowledge, our results for the first time demonstrated direct stimulation of the intestinal HIF-1 by prebiotic polysaccharides. This suggests that the microbiota-modulation effect of prebiotics not only depends on their fermentation by the gut bacteria, but also involves their direct interaction with the intestinal mucosa, which depends on the polysaccharide structure of the prebiotics as ligands. It is noteworthy that LGG EPS activated HIF1α by increasing mRNA transcription. This is in contrast to the hypoxic induction of HIF-1α, which relies on stabilization of HIF-1α protein. We found that TLR4ba signaling mediated the activation of HIF1 by LGG EPS, and the interaction of LGG EPS with TLR4ba was mediated by the main strand of the polysaccharide. Notably, while research suggests that zebrafish possess two TLR4 components (TLR4ba, TLR4bb), the tlr4ba and tlr4bb genes in zebrafish are paralogous rather than orthologous to human TLR4, and they fail to respond to LPS stimulation and subsequently trigger inflammation[63,64]. Previous studies have shown that the ERK (extracellular-signal-regulated kinase) p44/42 MAPK (ERK1/2)[65] and classical diacylglycerol–sensitive forms of protein kinase C (PKC)[66] are involved in the activation of HIF-1 by nonhypoxic stimuli. In this study, we observed that inhibition of p44/42 MAPK phosphorylation, but not PKCs, significantly reduced the LGG EPS-induced expression of HIF-1α mRNA (Supplementary Fig. 5a, b), suggesting that p44/42 MAPK signaling is the downstream pathway mediating the activation of HIF-1α by LGG EPS in zebrafish.

The regulation of intestinal microbial homeostasis by the HIF1α-AMP axis has been implicated previously in the context of alcoholic liver disease. However, the results were obtained by in vitro experiments, and no direct relationship between the

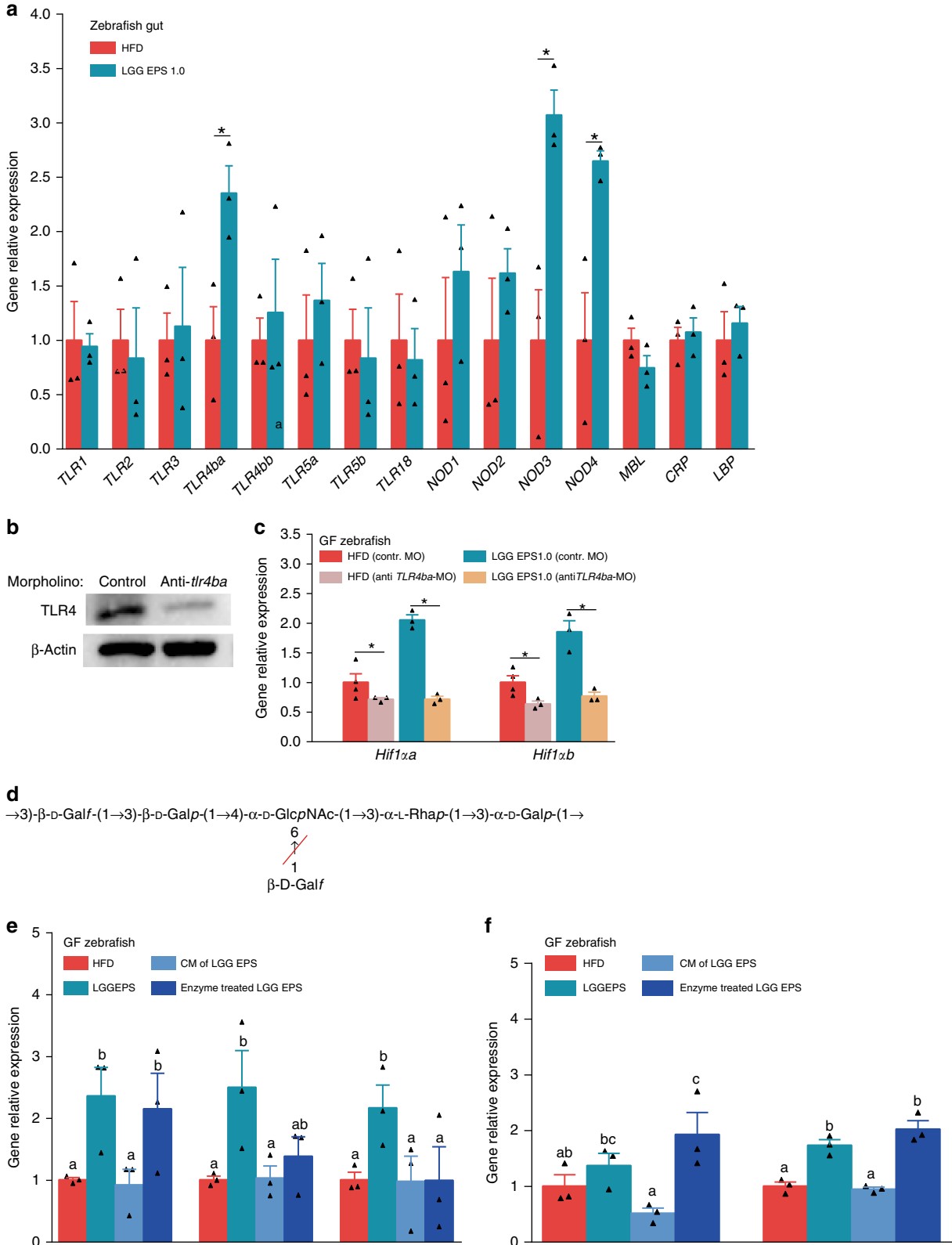

HIF1α-AMP pathway and the intestinal microbiota was established[17]. Our results show that HIF-1α activation regulates gut bacterial homeostasis through increasing production of antimicrobial peptides. The regulation of gut microbiota composition by AMPs has been reported previously[42,43]. Moreover, a previous study indicated that AMPs are the direct factor downstream of inflammasome signaling maintaining the homeostasis of the intestinal microbiota, and an aberrant AMP program by upstream inflammasome deficiency leads to microbial dysbiosis[44]. In this study, we observed that a combination of AMPs reversed the dysbiotic microbiota associated with BL23-EPS, indicating that the microbial homeostasis associated with LGG-EPS was due

**Fig. 8** The structural basis for the interaction between EPS and the receptors. **a** The expression of PRRs genes (*TLRs, NODs, MBL, CRP, LBP*) in the intestines of zebrafish fed on different diets for 4 weeks as measured by *q*-PCR (*n* = 3, pool of three zebrafish per sample). **b** Anti-*TLR4ba* vivo-morpholinos diminished TLR4 levels in zebrafish larvae. The expression of *Hif--1α* (**c**), in germ-free (GF) larvae fed sterile high-fat diet (HFD) or 1.0% LGG EPS diet and treated with vivo TLR4ba morpholino or control morpholino (*n* = 3 or 4, pool of 20 larvae per sample). **d** The structure of the repeating unit of the LGG EPS, and the target sites of β-Galactosidase (Red slash). The expression of PRRs genes (*TLR4ba, NOD3, NOD4*) (**e**), and *Hif-1α* (**f**), in the GF zebrafish fed with HFD, 1.0% LGG EPS diet, HF diet supplemented with 1.0% combination of monosaccharides comprised of LGG EPS or β-Galactosidase treated LGG EPS for 1 week (*n* = 3, pool of 20 larvae per sample). Data are expressed as the mean ± SEM. Graph bars labelled with different letters on top represent statistically significant results (*P* < 0.05), whereas bars with the same letter correspond to results that show no statistically significant differences

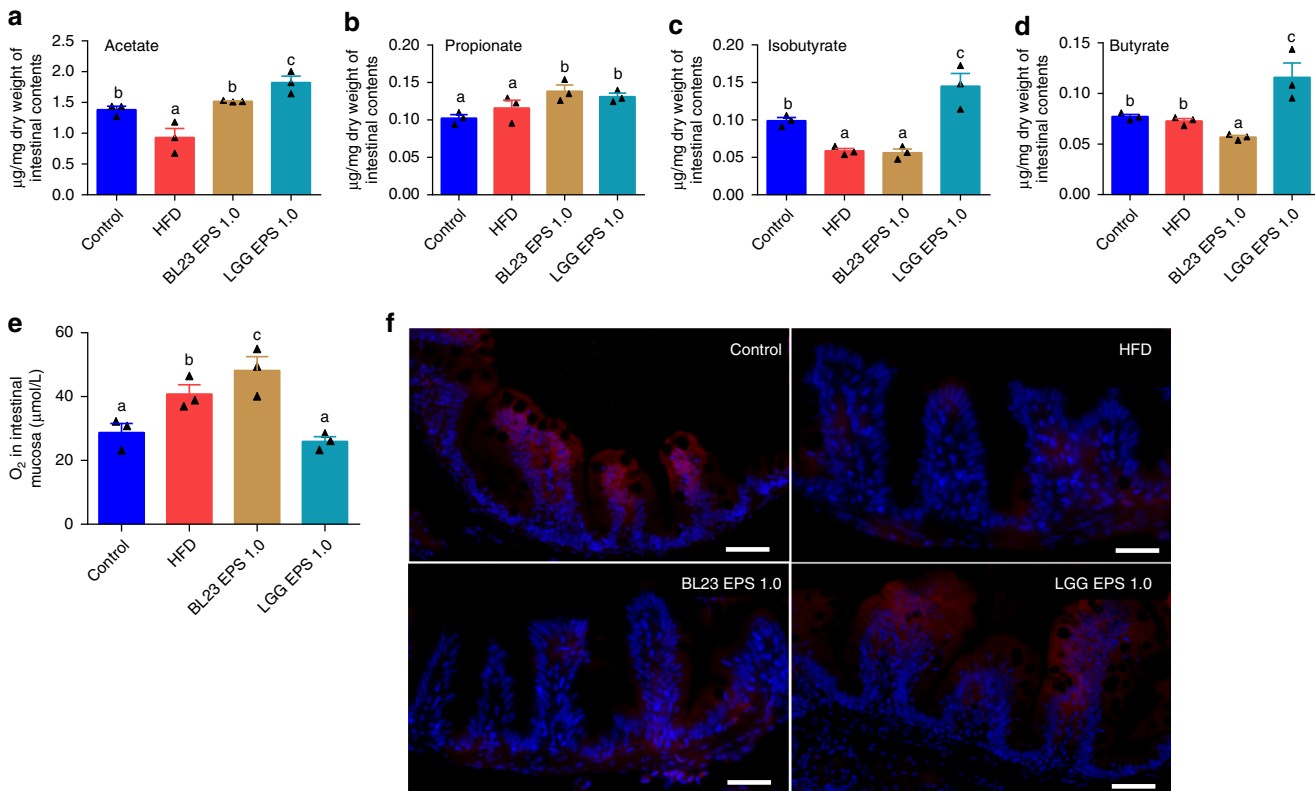

**Fig. 9** Butyrate further supported the differentiated state of microbiotas associated with LGG EPS or BL23 EPS. Adult zebrafish (1-month-old) were fed with the control diet, HF diet, or HF diet supplemented with 1.0% BL23 EPS or 1.0% LGG EPS for 4 weeks. Intestinal acetate levels (**a**), propionate levels (**b**), isobutyrate levels (**c**), butyrate levels (**d**) in zebrafish fed different diets which was performed with gas chromatography-mass spectrometry (*n* = 3). **e** The O₂ concentration in the intestinal mucosa layer of zebrafish fed with different diets for 4 weeks (*n* = 3). **f** Representative images of hypoxyprobe immunostaining in the intestine of zebrafish fed with different diets for 4 weeks. The scale bar is 50 μm. Data are expressed as the mean ± SEM. Graph bars labelled with different letters on top represent statistically significant results (*P* < 0.05), whereas bars with the same letter corresponds to results that show no statistically significant differences

to the enhanced AMP levels. Antimicrobial peptides defb1 and IL-37 have been reported to be positively regulated by HIF-1α[46,47]. Our results indicated that apart from defb1, defb2 and hepcidin were also positively correlated with HIF-1α, implying that they might also be regulated by HIF1α.

We observed lower partial oxygen pressure in the intestinal microenvironment of LGG-EPS fed zebrafish as compared with BL23-EPS group. The decreased oxygen level may be due to higher butyrate level in the intestinal content of LGG-EPS-fed zebrafish relative to the BL23-EPS group, as butyrate can decrease epithelial oxygenation[51]. The intestinal hypoxia induced by butyrate in LGG-EPS fed zebrafish may improve HIF1α protein stabilization, which contributes to the maintenance of intestinal microbiota composition associated with the LGG-EPS. Therefore, the metabolites of LGG-EPS-associated microbiota may further support the maintenance of the microbiota by activating the HIF1α-AMP axis. The supportive role of microbiota-metabolites to the maintenance of a homeostatic or dysbiotic microbiota has

been reported previously in the context of inflammasome sufficiency/deficiency[44], implying that this might be an important underlying scenario in the regulation of microbiota composition in different contexts. Furthermore, supplementation of butyrate in parallel to BL23 EPS ameliorated the dysbiosis (Supplementary Fig. 6), implying that butyrate may be used as an adjuvant of prebiotic polysaccharides to avoid the risk of inducing dysbiosis and liver injury.

Although BL23 EPS induced hepatic and intestinal inflammation (Fig. 3c, Supplementary Fig. 7a, b), it should be noted that the inflammation was induced by the microbiota. BL23 EPS does not induce inflammation directly. Rather it exhibited anti-inflammatory tendency when tested with zebrafish liver cells or germ-free zebrafish (Fig. 4a, Supplementary Fig. 1e). Our results indicated that the LGG EPS showed stronger anti-inflammatory effect than BL23 EPS *per se*. Interestingly, our results showed that the LGG EPS can inhibit the NFκB pathway, which may account for its anti-inflammatory effect. Moreover, this effect was

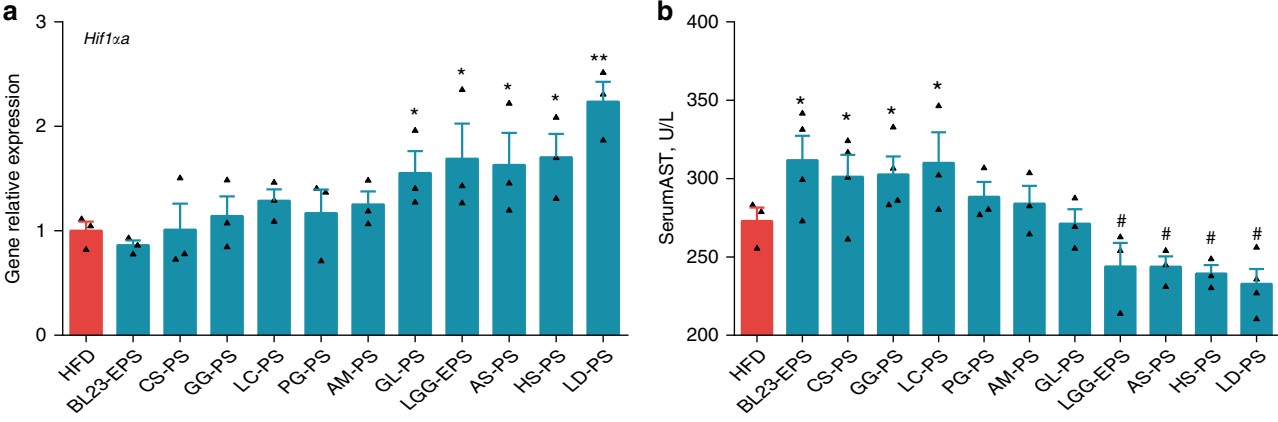

**Fig. 10** The liver injury risks of common polysaccharides based on HIF1α. Germ-free (GF) zebrafish larvae fed sterile high-fat diet (HFD) for 1 week and immersed in gnotobiotic zebrafish medium contained 10 μg/ml different common polysaccharides for another 3 days. The expression of *Hif-1αa* (**a**) in GF larvae (*n* = 3, pool of 20 larvae per sample). **b** Serum AST in adult zebrafish fed on HFD, or HFD supplemented with a polysaccharide at 1.0% for 4 weeks (*n* = 3 or 4, pool of three zebrafish per sample). Data are expressed as the mean ± SEM. Significant increase compared with HFD is designated as *P* < 0.05 (\*) and *P* < 0.01 (\*\*). Significant reduction compared with HFD is designated as *P* < 0.05 (#) and *P* < 0.01 (##)

independent of TLR4a, but probably involved NOD3 and NOD4 signaling (Supplementary Fig. 7c), which depends on the branch structure of the polysaccharide (Supplementary Fig. 7d).

Finally, we found that the liver injury risk was not confined to *Lactobacillus*-derived EPS, but can be broadly applicable to other types of commonly used prebiotic polysaccharides. Intriguingly, our data suggested that the liver injury risk of polysaccharides was correlated with the HIF1α activation effect. These results support the hypothesis that HIF1α-AMP axis acts as a key physiological force maintaining microbial homeostasis against the harmful fermentation associated with some risk-prone prebiotic polysaccharides in the context of a high-fat diet. The liver inflammation and injury effect of guar gum has been reported in mice[29]. The consistency of phenotypes of guar gum in zebrafish and mice indicates that our results are not due to the specificity of the zebrafish host, and may be extrapolated to mammals. The unexpected adverse effect of prebiotic polysaccharides (fibers) has been reported recently[30]. The findings in this study may provide some insight in the mechanisms underlying the risk of prebiotic polysaccharides, and suggest that the HIF1α-AMP axis may be harnessed as a target to promote safe use of prebiotics.

## Methods

**EPS extraction and purification.** *L. rhamnosus* ATCC53103 (LGG) and *Lactobacillus casei* BL23 were purchased from the China Center of Industrial Culture Collection (Beijing, China) and stationarily cultivated in de Man, Rogosa and Sharpe (MRS) broth containing 10 g/L of tryptone, 10 g/L of beef extract, 5 g/L of yeast extract, 10 g/L of glucose, 5 g/L of sodiumacetate·3H₂O, 1 g/L of Tween-80, 2 g/L of citric acid ammonium salt dibasic, 0.2 g/L of MgSO4·7H₂O, 0.05 g/L of MnSO₄·H₂O, and 2 g/L of K₂HPO₄) at 37 °C for 48 h. Then, the bacterial cells were harvested in a refrigerated centrifuge (12,000 rpm for 10 min at 4 °C) and washed thrice with distilled water to remove the MRS broth.

Extraction and purification of LGG EPS was conducted as previously reported[67]. Briefly, total EPS were extracted from the washed lactobacilli cells by mild sonication (40 W, 10 min) in a water bath (80 °C) for 4 h. The EPS were precipitated by gradually adding cold ethanol to 75% (v/v) to the filtered supernatant. The precipitated EPS material was obtained by centrifugation, washed, and dissolved in water obtained from an Alpha-Q reagent grade water purification system (Millipore Co., Milford, MA, USA). The aqueous solution of the EPS was further treated with Sevag reagent at a final concentration of 25% and incubated for 2 h under gentle agitation. The precipitated proteins were removed by centrifugation at 5,000 *g* for 20 min. After centrifugation, the solution containing EPS was dialyzed (molecular weight cut-off: 10,000 Da) against 5 L of distilled water for 2 days with three water changes per day. The EPS (8 mg/mL) was further purified by size-exclusion chromatography (SEC) on a column of Superdex75 (10/300 GE) (Pharmacia, Uppsala, Sweden) fitted to an AKTA FPLC system (Pharmacia) and eluted with 0.3 M NaCl (sodium chloride) solution. The amount

of carbohydrate was estimated by the phenol-sulfuric acid method. The EPS material eluted in the void volume was lyophilized.

**Zebrafish liver cell culture and treatments.** Zebrafish liver cells (ATCC number CRL-2643) were cultured in LDF culture medium (50% Leibovitz's L-15, 35% Dulbecco's modified Eagle's and 15% Ham's F12 media) supplemented with 5% fetal bovine serum, 0.5% trout serum, 10 μg/mL bovine insulin, 50 ng/mL mouse epidermal growth factor, and penicillin/streptomycin. The cells were maintained at 28 °C in a humidified 5% CO₂ atmosphere. Upon reaching confluency, the cells were treated with 100 μM oleic acid (OA) for 24 h to establish a hepatocyte fatty degeneration model. This model is used to test the effects of EPS on the cellular levels. The cells were treated with EPS (10 μg/mL) or equal volume of ddH₂O after the cells covered the plate, and cells were harvested 24 h after treatment.

**Zebrafish husbandry and experimental diets.** All experimental and animal care procedures were approved by the Feed Research Institute of the Chinese Academy of Agricultural Sciences Animal Care Committee, under the auspices of the China Council for Animal Care (assurance No. 2016-AF-FRI-CAAS-001). Fatty liver models were established by feeding the fish with a high-fat diet. Lard oil and soybean oil were added to replace an equal quantity of dextrin in the basal diet to make the high-fat diet (isonitrogenous diet). The content of crude fat of the high-fat diet and the basal diet (control) for adult fish was 16% and 6%, respectively (Supplementary Table 1), while it was 20% and 9%, respectively, for the larval zebrafish (Supplementary Table 2).

Adult zebrafish (1-month-old) were fed with the experimental diets twice a day (9:00, 17:00) to apparent satiation each time for 4 weeks. Adult zebrafish were randomly assigned to 2-L tanks in a recirculating system with 18 fish in each tank. Embryos and larvae were reared in embryo medium at 28 °C to 4 day-postfertilization (dpf). Zebrafish larvae at 4 dpf were allocated randomly to tanks with 100 mL of water and 80 larvae per tank. The larvae started to feed at 5 dpf for 7 days. They were fed with the experimental diets twice a day to apparent satiation each time. Water was replaced by half every day. For EPS supplemention, the high-fat diet for adult and larval zebrafish were mixed with 0.5% or 1.0% EPS, which replaced the same amount of zeolite, and then ground to properly-sized particles (Supplementary Tables 1 and 2).

**Histology, immunohistochemistry, and oil Red O staining.** For hematoxylin and eosin (H&E) staining and TUNEL staining, zebrafish liver, intestine, or whole zebrafish larvae were rinsed with sterilized PBS, fixed in 4% paraformaldehyde in PBS, and then embedded in paraffin for H&E and TUNEL staining. For immunohistochemistry, formalin-fixed and paraffin-embedded sections were blocked with endogenous peroxidase (3% H₂O₂ in 80% methanol) for 20 min. Antigen retrieval was performed in 10 mM sodium citrate in a microwave for 15 min. After blocking nonspecific antigen with normal goat serum for 30 min, the slides were incubated with C/EBPα (Bioworld, BS1384, 1:200 dilution), DGAT2 (Bioworld, BS60142, 1:300 dilution), PPARα (Bioworld, BS1689, 1:200 dilution), or CPT1 (Bioworld, BS7744, 1:300 dilution) antibody overnight at 4 °C. The slides were then incubated with biotinylated-labelled secondary antibodies (1:200, GE Health, UK) for 30 min at room temperature. Visualization was performed using 0.1% 3,3'-diaminobenzidine (Dako, Denmark) in PBS together with 0.05% H₂O₂.

For Oil Red O staining of liver, liver sections (10 μm thick) were washed with PBS and then fixed with 4% paraformaldehyde in PBS for 1 h at room temperature.

The samples were washed with PBS, stained with a filtered Oil Red O (ORO; Sigma-Aldrich) stock solution (0.5 g ORO in 100 mL of isopropyl alcohol) for 15 min at room temperature, washed with deionized water, counterstained using H&E, and then mounted. For ORO whole-mount analysis, whole larvae were fixed with 4% paraformaldehyde, washed with PBS, infiltrated with a graded series of propylene glycol baths, and stained with 0.5% ORO in 100% propylene glycol overnight. The stained larvae were washed with decreasing concentrations of propylene glycol, followed by several rinses with PBS, and stored in 80% propylene glycol bath. Larvae were defined positive for steatosis when the boundary between the liver and surrounding tissue is clear. Images were obtained using a microscope (Leica DMIL-LED, Germany). The images were converted to 8-bit gray scale for measuring mean gray value using ImageJ software. The intensity of images was quantified using ImageJ and used to quantitatively evaluate fatty liver hepatocyte steatosis.

**Triacylgycerol assay**. Tissues were homogenized in PBS buffer with protease inhibitors. A chloroform/methanol (2:1) solution was rapidly added to the homogenate, and the samples were vortexed. The samples were centrifuged at 250 g for 10 min to separate the phases. The lower lipid-containing phase was carefully aspirated and allowed to dry in a 70 °C metal bath with nitrogen steam. The dried lipids were emulsified in chloroform with 5% Triton X-100. Finally, the dried emulsified lipids with nitrogen gas were reconstituted in distilled water. Triacylglyceride content was measured by enzymatic reaction according to the instruction manual (Wako Diagnostics, Japan). For cellular triacylglyceride level, lipids were extracted from zebrafish liver, and dissolved in isopropanol. Triacylglyceride content was normalized to the protein amount of each sample.

**Tissue and serum biochemical measurements**. Blood samples were collected from zebrafish as previously described[68]. Serum ALT and aspartate aminotransferase AST activities were detected using commercial diagnostic kits (Nanjing Jiancheng Bioengineering Institute, Jiangsu Province, China) according to the manufacturer's instructions. Serum ALT and AST activity was examined at 510 nm according to the manufacturer's instructions, and was expressed as enzyme activity units per liter (U/L). LPS level was determined using the ToxinSensorTM Chromogenic LAL Endotoxin Assay Kit (Genscript, Jiangsu Province, China) according to the manufacturer's instructions. The serum level of LPS in adult zebrafish was expressed as LPS units per milliliter (EU/mL). The hepatic level of LPS in adult zebrafish was expressed as LPS units per mg liver (EU/g).

**Germ-free zebrafish husbandry and gut microbiota transfer**. Germ-free zebrafish were prepared following established protocols as described previously[69,70]. To determine the direct effect of EPS on germ-free zebrafish, diets for larval zebrafish were sterilized by irradiation with 20 kGy gamma ray (Institute of Food Science and Technology, Chinese Academy of Agricultural Sciences, Beijing, China). Zebrafish larvae hatched from their chorions at 3 dpf, and the yolk was largely absorbed at 5 dpf. Germ-free zebrafish started to feed at 5 dpf for seven days. After one week of feeding, the zebrafish were fasted for 12 h before sample collection for RNA extraction, and for 24 h before sample collection for whole-mount ORO staining. The transfer of gut microbiota to germ-free zebrafish was performed according to Rawls et al.[70] with minor modifications[69,71]. The gut microbiota was added to gnotobiotic zebrafish medium (GZM) containing 3 dpf germ-free zebrafish at a concentration of $10^6$ CFUs/mL GZM. At 5 dpf, the zebrafish recipients were fed high-fat diet for seven days. The transfer efficiency was confirmed by DGGE as described elsewhere[69]. After 1 week of feeding, the zebrafish were fasted for 12 h followed by sample collection for RNA extraction, and for 24 h before sample collection for whole-mount Oil Red O staining.

**Quantitative PCR analysis**. Total RNA was isolated from zebrafish liver and zebrafish samples using Trizol (Invitrogen). First-strand complementary DNA synthesis was performed using the Superscript First-Strand Synthesis System. Quantitative real-time PCR reaction was performed using the SYBR Green Supermix (TianGen, China) on a Light Cycler 480 system (Roche). The primer sequences are listed in Supplementary Table 6.

**Western blotting**. Zebrafish intestines or larval zebrafish were lysed with ice-cold RIPA lysis buffer mixed with 1 mM PMSF and phosphatase inhibitors (Abcam, USA). Equivalent amounts of total protein were loaded into a 12% SDS-PAGE for electrophoresis and then transferred onto a polyvinylidene difluoride (PVDF) membrane (Millipore, USA). After blocking nonspecific binding with 5% non-fat dry milk in PBS, the PVDF membrane was incubated with primary antibodies, i.e., antibodies against β-actin (CMCTAG, AT0544, 1:1000), HIF-1α (Bioworld, BS3514, 1:1000), TLR4 (Cell Signaling Technology, 14358S, 1:1000). The blots were developed using horseradish peroxidase (HRP)-conjugated secondary antibodies (GE Health, 1:3000) and the ECL-plus system.

**Gut microbiota analysis**. At the end of the 4-weeks feeding period, the digesta of adult zebrafish were collected 4 h after the last feeding. The digesta were collected under aseptic conditions. The digesta samples from the 6 fish were pooled as a

replicate. DNA was extracted from each pooled sample using a Fast DNA SPIN Kit for Soil (MP Biomedicals), according to the manufacturer's instructions. The 16s V3–V4 region was amplified by using the primers U341F (5′-CGGCAAC GAGCGCAACCC-3′) and U806 (5′-CCATTGTAGCACGTGTGTAGCC-3′). 16s rRNA gene sequencing was performed at the Realbio Genomics Institute (Shanghai, China) using the Illumina HiSeq platform. Microbiota sequencing data in this study are available from the European Nucleotide Archive under accession number PRJEB25167.

The number of total bacteria or a specific phylotype was quantified by q-PCR. Primer sets for universal bacteria or specific bacterial groups targeted the 16S rRNA gene and are listed in Supplementary Table 6. 16S rRNA of each bacterial strain was cloned into the pLB vector (Tiangen, Beijing, China) according to the manufacturer's procedure as a copy number standard. For each q-PCR standard, the copy number concentration was calculated based on the length of the PCR product and the average mass of a DNA base pair. For the adult zebrafish, results were expressed as Log10 copy numbers of bacterial 16S rDNA per milligram of intestinal contents. For the larval zebrafish, results were expressed as Log10 copy numbers of bacterial 16S rDNA per larva. For the gut microbiota cultured in vitro, results were expressed as Log10 copy numbers of bacterial 16S rDNA per ml medium.

**Morpholino knockdown**. Vivo-morpholino oligonucleotides (MO) against zebrafish *tlr4ba* were designed and synthesized by Gene-Tools (Philomath, OR). The sequences of MO used in this study are as follows: *tlr4ba* MO (translating blocking), 5′-GATGCTGCTGAGGTTTCTTCCCATG-3′; and standard control MO, 5′-CCTCTTACCTCAGTTACAATTTATA-3′. When applying the MO, zebrafish husbandry and all experimental procedures were the same as earlier described, except for GZM that contained 100 nmol of *tlr4ba* MO, or standard control MO during the entire 1-week treatment[72,73].

**Short chain fatty acid analysis**. Gut content and bacteria culture medium were collected for short chain fatty acid analysis. Gut content samples were collected from zebrafish 4 h post the last feeding. The gut contents from 5–6 fish were pooled. Gut content sample or 1 mL of bacteria culture medium was lyophilized and resuspended with 0.2 or 1 mL of MeOH, respectively. Samples were mixed vigorously with sonication for three times with 10 min for each. After sonication, the samples were centrifuged at 12000 rpm for 10 mins, and the supernatants were used for GC−MS analysis. GC−MS was performed on a GCMS-QP2010 Ultra with an autosampler (SHIMADZU) and the Rtx-wax capillary column (60 m, 0.25 mm i.d., 0.25 μm film thickness; SHIMADZU). Oven temperature was programmed from 60 to 100 °C at 5 °C/min, with a 1 min hold; to 150 °C at 5 °C/min, with a 5 min hold; to 225 °C at 30 °C/min, with a 20 min hold. Injection of a 2 μL sample was performed at 230 °C. Helium, at a flow of 1.2 mL·min$^{-1}$, was the carrier gas. Electronic impact was recorded at 70 eV. The weight of each lyophilized gut content sample was recorded for calibration.

**Hypoxyprobe staining**. Zebrafish fed for 4 weeks were injected with 10 μL of a 10 mg/mL pimonidazole solution (HP7; Hypoxyprobe) for 3 days. Then zebrafish were anesthetized with tricaine methanesulfonate (MS222) for intestine isolation. The isolated intestine was fixed in 4% PFA for immunohistochemical analysis. Sections were blocked in endogenous peroxidase (3% $H_2O_2$ in 80% methanol) for 20 min. Antigen retrieval was performed in 10 mM sodium citrate in a microwave for 15 min. After blocking nonspecific antigen with normal goat serum for 30 min, the slides were incubated with a primary fluorescein (Dylight549)-conjugated mouse monoclonal antibody (Mab) directed against pimonidazole protein adducts antibody overnight at 4 °C. Then the slides were incubated with a secondary goat anti-mouse Alexa-Fluor 647 (A-21235; Life Technologies) for 30 min at room temperature[74].

**Intestinal $O_2$ measurements using microelectrodes**. A set of three zebrafish was used to measure the profiles of $O_2$ at each midgut. Oxygen microelectrodes (OX-25; Unisense, Aarhus, Denmark) were used for the measurement of $O_2$ concentration as previously described[75]. Before use, the electrodes were polarized and calibrated in water saturated with air, as well as in saturated $Na_2SO_3$ solution (zero oxygen concentration). Before microelectrode measurements, 50 ml of low melting point agarose consisting of 1% agarose in PBS was cast into a microchamber. A freshly dissected gut was placed on this layer of agarose, fully extended and immediately covered with a second layer of molten agarose at 30 °C. Measurements were performed radially starting at the surface of the gut wall (0 μm) through the zebrafish gut until the tip completely penetrated the whole tissue. All measurements were carried out at room temperature (25 °C).

**Statistics and Reproducibility**. The statistical data reported include results from at least three biological replicates. All results are expressed as the mean ± SEM. All statistical analyses were performed in GraphPad Prism Version 6 (GraphPad Software). Comparisons between two groups were analyzed using Student's *t*-test, and comparisons between multiple groups were analyzed using one-way ANOVA followed by a Duncan's test. Differences were considered significant at $P < 0.05$ (*) and $P < 0.01$ (**).

**Reporting summary**. Further information on research design is available in the Nature Research Reporting Summary linked to this article.

## Data availability

Microbiota sequencing data in this study are available from the European Nucleotide Archive (ENA) under accession number PRJEB25167. The source data for all figures are provided as Supplementary Data 1. The authors declare that all other data supporting the findings of this study are available within the article and its supplementary information files.

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

## Acknowledgements

National Key R&D Program of China (2018YFD0900400) and the National Natural Science Foundation of China (NSFC 31802315, 31760762, 31872584 and 31702354).

## Author contributions

Z.Z.G. designed the research. Z.Z. and R.C. wrote the paper, and Z.Z.G. gave conceptual advice for the paper. Z.Z. performed experiments and acquired data. D.Q.W., G.C.C. and L.H.L. assisted in western blot experiments, the immunofluorescence staining experiments, and the morpholino knockdown experiments. D.Q.W., L.H.L. and X.Y.D. participated in zebrafish sampling. Y.Y.L. and Z.H.L. co-analyzed and discussed the results. All authors have read, commented on, and approved the final paper.

## Additional information

**Competing interests:** The authors declare no competing interests.

