## [Peer Review File · Communications Biology]

Reviewers' comments:

Reviewer #1 (Remarks to the Author):

Zhang et al utilize a zebrafish model of fatty liver disease to demonstrate contrasting effects of exopolysaccharides (EPS) from two strains of *Lactobacillus*. They provide evidence that LGG EPS induces HIF1a activity, leading to increased antimicrobial production and microbial homeostasis. BL23 EPS, in contrast, induced intestinal dysbiosis with subsequent increased hepatic inflammation and apoptosis.

Overall, I believe it is a well-designed study to examine the effects of probiotic-derived polysaccharides on the microbiome and steatohepatitis. It provides novel insights that I anticipate would influence thinking in this particular area of research. The authors show that the effects of BL23 EPS are microbe mediated using germ-free and antibiotic-treated models, as well as germ-free zebrafish colonized with microbiota of donor zebrafish receiving the various EPS. The authors provide evidence from a foxo3b KO model and a HIF inhibitor model to support their hypothesis that the differences between the two EPS reflects differences in HIF1 signaling, TLR signaling and antimicrobial induction between the two EPS. Experimental methodology is described in sufficient detail to reproduce the work and statistical analyses are valid.

Comments/Questions:

- 1) In their HIF1a knockdown model with YC-1 combined with LGG EPS (which caused intestinal dysbiosis similar to BL23 EPS), did the authors see elevated AST/ALT and hepatic apoptotic markers similar to the BL23 EPS supplemented zebrafish? This would further consolidate their hypothesis that dysbiosis is a critical mediator of hepatic inflammation and that the protective role of LGG EPS is mediated by HIF1a-dependent microbial homeostasis.
2. Can the authors show metrics of alpha diversity (such as chao1 or shannon index) in response to EPS? Dysbiosis is often characterized by reduction in alpha diversity metrics.
3. In addition to showing the differential abundance of specific genera, as the authors did, can the authors show beta-diversity differences across the samples via a principal coordinates analysis (PCoA) plot?
4. The authors associate BL23 EPS inflammation with the inflammation seen in both NASH and alcoholic liver disease models. However, in both these models, severity of steatosis and hepatitis typically increase in parallel. BL23 EPS seems to improve steatosis but induce inflammation irrespective of the reduced level of steatosis. The authors never comment on why they think BL23 EPS would have these seemingly contrasting effects on the liver.
5. Careful proofreading is required to correct typographical errors, misspellings, and incorrect labeling that are scattered throughout the manuscript.

Reviewer #2 (Remarks to the Author):

The study by Zhang et al. indicated the anti-hepatosteatosis effect of natural polysaccharides in HFD model and they found the natural polysaccharides BL23 EPS, but not LGG EPS induced liver inflammation and injury. The authors aimed to explore the detailed mechanism for BL23 EPS induced liver inflammation and injury. The authors report that LGG EPS, but not BL23 EPS, could activate HIF1a-AMP axis to regulate the gut microbial composition and decrease LPS level, LBP expression level in liver. The LPS and LBP could promote a signaling cascade in the liver and induced inflammation and injury. Although these findings that LGG EPS increase AMP level to facilitate the microbial homeostasis, are of significance, the study appears contains several concerns.

1. In the Figure10, the authors stated that "the liver injury risk observed in our study may be extrapolated to other commonly used prebiotic polysaccharides, and it is correlated with the HIF1a

activation efficiency of the polysaccharides." They only selected 4 PS to verify their hypothesis. Is the selection processed randomly? I think more PS should be included to verify the hypothesis.

2. The image of liver histological changes was not very convincing. The quantification score of immunohistochemistry staining should be assessed.

3. For FMT experiment, authors should provide data of the intestinal microbiota from control, HFD, or HFD containing LGG EPS or BL23 EPS group donor/recipient to show the microbiota was successfully colonized.

4. In the lines 274-275, authors indicated the enhanced expression of TLR1 and LBP in the liver of BL23 EPS treated zebrafish and they considered LBP signaling mediated the liver injury effect.

However, the evidence is not fully supportive of such a conclusion. Therefore, the biological role of LBP signaling in the BL23 EPS induced liver injury should be investigated, for example, the downstream inflammatory signaling pathways activation.

5. beta-diversity, specifically for PCoA analysis is needed for the dysbiosis part.

Referee expertise:

Referee #1: Gut microbiota and NAFLD

Referee #2: Gut dysbiosis and liver disease

Reviewers' comments:

Reviewer #1 (Remarks to the Author):

Zhang et al utilize a zebrafish model of fatty liver disease to demonstrate contrasting effects of exopolysaccharides (EPS) from two strains of *Lactobacillus*. They provide evidence that LGG EPS induces HIF1a activity, leading to increased antimicrobial production and microbial homeostasis. BL23 EPS, in contrast, induced intestinal dysbiosis with subsequent increased hepatic inflammation and apoptosis.

Overall, I believe it is a well-designed study to examine the effects of probiotic-derived polysaccharides on the microbiome and steatohepatitis. It provides novel insights that I anticipate would influence thinking in this particular area of research. The authors show that the effects of BL23 EPS are microbe mediated using germ-free and antibiotic-treated models, as well as germ-free zebrafish colonized with microbiota of donor zebrafish receiving the various EPS. The authors provide evidence from a *foxo3b* KO model and a HIF inhibitor model to support their hypothesis that the differences between the two EPS reflects differences in HIF1 signaling, TLR signaling and antimicrobial induction between the two EPS. Experimental methodology is described in sufficient detail to reproduce the work and statistical analyses are valid.

Comments/Questions:

1. In their HIF1a knockdown model with YC-1 combined with LGG EPS (which caused intestinal dysbiosis similar to BL23 EPS), did the authors see elevated AST/ALT and hepatic apoptotic markers similar to the BL23 EPS supplemented zebrafish? This would further consolidate their hypothesis that dysbiosis is a critical mediator of hepatic inflammation and that the protective role of LGG EPS is mediated by HIF1a-dependent microbial homeostasis.

Response: Thank you for your advice. We have supplemented the suggested results.

YC-1 treatment elevated the level of ALT (Fig. 7m) and AST (Fig. 7n) and increased the expression of pro-apoptotic factors (*Noxa* and *Puma*) (Fig. 7o) in the liver of the LGG EPS treated zebrafish.

2. Can the authors show metrics of alpha diversity (such as chao1 or shannon index) in response to EPS? Dysbiosis is often characterized by reduction in alpha diversity metrics.

Response: The alpha diversity metrics have been added in Supplementary Table 1. The Chao1 and Shannon index indicate that HFD group has higher alpha diversity than control, while BL23 EPS treatment lowered Chao1 and increased Shannon index compared with HFD group. In contrast, LGG EPS group showed the lowest alpha diversity, as reflected by all the indexes. This is not consistent with what we usually observe from a dysbiotic microbiota versus an homeostatic/control one. However, as reviewed by Hooks and O'Malley, dysbiosis can be categorized into three categories, including general change in the microbiota composition, an imbalance in composition, and changes to specific lineages in that composition, and dysbiosis is not always accompanied by reduction in alpha diversity (Hooks and O'Malley, *mBio* 2017). The BL23

EPS microbiota showed significant change in the microbiota structure as relative to the control and HFD group, and featured an expansion of the *Proteobacteria* and reduction of *Fusobacteria*, which met the criteria of dysbiosis as summarized by Hooks and O'Malley.

Supplementary Table 1. Diversity index of gut bacteria of zebrafish fed with control, HFD, BL23 EPS or LGG EPS-supplemented diet for four weeks¹.

Sample	OTUs	Chao1	PD whole tree	Simpson	Shannon
Control	164.3 ± 18.8 ^b	157.9 ± 13.6 ^b	15.9 ± 1.5 ^b	0.67 ± 0.03 ^b	2.7 ± 0.3 ^b
HFD	220.3 ± 28.2 ^c	249.3 ± 25.6 ^c	19.6 ± 2.3 ^c	0.84 ± 0.03 ^c	3.9 ± 0.3 ^c
BL23EPS 1.0	140.5±20.8 ^{ab}	137.9±20.5 ^{ab}	14.3±1.5 ^{ab}	0.90±0.01 ^c	4.2 ± 0.1 ^c
LGGEPS 1.0	93.0 ± 21.6 ^a	86.3± 22.7 ^a	9.1 ± 2.6 ^a	0.50 ± 0.10 ^a	1.6 ± 0.4 ^a

¹Values are expressed as the mean ± SEM, n = 3 or 4. Chao1, Chao1 index; OTU, operational taxonomic unit; PD, phylogenetic diversity; Simpson, Simpson's diversity index; Shannon, Shannon diversity index. Means marked with different letters represent statistically significant results ($P < 0.05$), whereas the same letter correspond to results that show no statistically significant differences.

3. In addition to showing the differential abundance of specific genera, as the authors did, can the authors show beta-diversity differences across the samples via a principal coordinates analysis (PCoA) plot?

Response: The weighted version of UniFrac-based PCoA analysis result has been added. From the PCoA figure we can see that the BL23 EPS associated microbiota is different from the other three groups, especially on PCoA1(Supplementary Fig. 3b). Further, the PCA analysis of the genera with different abundance among groups was conducted (Supplementary Fig. 3c). The result showed that while LGG EPS associated microbiota resembles that of the control group, the BL23 EPS-microbiota deviates apart from all the other three groups, which is overall consistent with the abundance data.

4. The authors associate BL23 EPS inflammation with the inflammation seen in both NASH and alcoholic liver disease models. However, in both these models, severity of steatosis and hepatitis typically increase in parallel. BL23 EPS seems to improve steatosis but induce inflammation irrespective of the reduced level of steatosis. The authors never comment on why they think BL23 EPS would have these seemingly contrasting effects on the liver.

Response: Yes BL23 EPS reduced steatosis but induced liver inflammation and injury, which are seemingly contrasting effects in the liver. We associate BL23 EPS inflammation with that of NASH and ALD as all of them involved gut microbial dysbiosis and translocation of microbial products from the intestinal lumen. We now added some comments on the typical phenotype of BL23 EPS (Line 533-539). Similar with BL23-EPS, guar gum also induced liver inflammation and injury while improving hepatic steatosis (Janssen et al., Journal of Lipid Research, 2017). The phenotypes associated with BL23 EPS and guar gum indicate that hepatic inflammation and injury may occur independent of the steatosis in some conditions, suggesting a multifactorial etiology of this symptom.

5. Careful proofreading is required to correct typographical errors, misspellings, and incorrect labeling that are scattered throughout the manuscript.

Response: Thank you for your detailed review. Careful proofreading has been made throughout the manuscript.

Reviewer #2 (Remarks to the Author):

The study by Zhang et al. indicated the anti-hepatosteatosis effect of natural polysaccharides in HFD model and they found the natural polysaccharides BL23 EPS, but not LGG EPS induced liver inflammation and injury. The authors aimed to explore the detailed mechanism for BL23 EPS induced liver inflammation and injury. The authors report that LGG EPS, but not BL23 EPS, could activate HIF1 α -AMP axis to regulate the gut microbial composition and decrease LPS level, LBP expression level in liver. The LPS and LBP could promote a signaling cascade in the liver and induced inflammation and injury. Although these findings that LGG EPS increase AMP level to facilitate the microbial homeostasis, are of significance, the study appears contains several concerns.

1. In the Figure10, the authors stated that “the liver injury risk observed in our study may be extrapolated to other commonly used prebiotic polysaccharides, and it is correlated with the HIF1 α activation efficiency of the polysaccharides.” They only selected 4 PS to verify their hypothesis. Is the selection processed randomly? I think more PS should be included to verify the hypothesis.

Response: Thank you for your suggestion. we collected 9 natural polysaccharides with reported anti-steatosis effect. We randomly chosen 4 of them in our previous experiment. To further verify our hypothesis, we complemented experiment and tested the liver injury risk of all the 9 PS. Serum AST level was tested as the marker for liver injury instead of LPS level. The results showed that the 5 PS with no HIF1 α activation activity all led to increased serum AST level after 4 weeks of feeding, while the 4 PS that can stimulate HIF1 α led to unchanged (for GL-PS) or significantly decreased levels of serum AST (for the other three PS) compared with HFD group (Fig. 10a, b)

Figure 10. The liver injury risks of common polysaccharides based on HIF1 α . GF zebrafish larvae fed sterile HFD for one week and immersed in gnotobiotic zebrafish medium contained 10 μ g/ml different common polysaccharides for another 3 days. The expression of *Hif-1 α* (a) in GF larvae (n = 4, pool of 20 larvae per sample). (b) Serum AST in adult zebrafish fed on HFD, or HFD supplemented with a polysaccharide at 1.0% for four weeks (n = 3, pool of 3 zebrafish per sample). Data are expressed as the mean \pm SEM. Significant increase compared with HFD is designated as $P < 0.05$ (*) and $P < 0.01$ (**). Significant reduction compared with HFD is designated as $P < 0.05$ (#) and $P < 0.01$ (##) .

2. The image of liver histological changes was not very convincing. The quantification score of immunohistochemistry staining should be assessed.

Response: The quantification score of the liver histological images has been assessed with mean optical density (MOD) by using Image-pro plus 6.0 (Media Cybernetics, Inc., Rockville, MD, USA). Each group includes three replicates. The results are as follows:

3. For FMT experiment, authors should provide data of the intestinal microbiota from control, HFD, or HFD containing LGG EPS or BL23 EPS group donor/recipient to show the microbiota was successfully colonized.

Response: Thank you for your suggestion. This is a very important point. The microbiota transfer technique has been established in our lab for several years, following the methods in the literature with some modifications (Rawls et al., *PNAS* 2004). To confirm the successful colonization of the microbiota, we analyzed the microbiota colonized in the GF zebrafish and comparing it with the original microbiota from the donor fish. The results showed a high similarity between the donor and recipient microbiotas. This has been published in our previous study (Ran et al., *Journal of Nutrition* 2016; Guo et al., *Journal of Nutrition* 2017;).

4. In the lines 274-275, authors indicated the enhanced expression of TLR1 and LBP in the liver of BL23 EPS treated zebrafish and they considered LBP signaling mediated the liver injury effect. However, the evidence is not fully supportive of such a conclusion. Therefore, the biological role of LBP signaling in the BL23 EPS induced liver injury should be investigated, for example, the downstream inflammatory signaling pathways activation.

Response: LBP is a soluble receptor that binds LPS. *LBP* expression was induced in the liver of BL23 EPS treated fish but not by BL23 EPS treatment in ZFL cells and GF zebrafish (Supplementary Fig3. b-d), suggesting that LBP was induced by LPS translocated from the intestinal lumen of BL23 EPS treated zebrafish, and that the translocated LPS contributed to the liver injury effect of BL23 EPS. Supporting this, we observed that the LPS level in the serum and liver of BL23 EPS fed zebrafish was significantly higher than HFD fish (Fig. 5d,e). The LBP is a soluble LPS receptor that presents LPS to the receptors on the cell surface. The TLR for LPS recognition in fish is still unknown. However, previous studies have indicated that LPS can induce inflammation in zebrafish (Sullivan et al., *Journal of Immunology* 2009; Sepulcre et al., *Journal of Immunology* 2009; Yang et al., *Nature Communications* 2018). Based on the induced LBP expression and enhanced LPS level in the serum and liver, we propose that LPS translocated from intestinal lumen contributed to the liver inflammation and injury effect of BL23 EPS. The signaling pathways will be investigated in further studies. This part of text

has been revised to make it more clear (Line 274-282), and the term “LBP signaling” has been deleted, as it is a little bit misleading.

5. *beta-diversity, specifically for PCoA analysis is needed for the dysbiosis part.*

Response: The weighted version of UniFrac-based PCoA analysis result has been added. From the PCoA figure we can see that the BL23 EPS associated microbiota is different from the other three groups, especially on PCoA1 (Supplementary Fig. 3b). Further, the PCA analysis of the genera with different abundance among groups was conducted (Supplementary Fig. 3c). The result showed that while LGG EPS associated microbiota resembles that of the control group, the BL23 EPS-microbiota deviates apart from all the other three groups, which is overall consistent with the abundance data.

REVIEWERS' COMMENTS:

Reviewer #1 (Remarks to the Author):

The authors have adequately addressed each of my concerns within their paper. They have also added key figures that further strengthen their findings.

Reviewer #2 (Remarks to the Author):

No more concerns